



# Ozone causes substantial reductions in the carbon sequestration of managed European forests

Per Erik Karlsson[1], Patrick Büker[2], Sam Bland[3], David Simpson[4], Katrina Sharps[5], Felicity Hayes[5], Lisa D. Emberson[6].

[1]IVL Swedish Environmental Research Institute, P.O. Box 53021, SE-40014 Gothenburg, Sweden
[2]Deutsche Gesellschaft für Internationale Zusammenarbeit GmbH, Bonn 53113, Germany
[3]Stockholm Environment Institute at York, University of York, U.K.
[4]EMEP MSC-W, Climate Modelling and Air Pollution Division, Norwegian Meteorological Institute, Oslo, Norway
[5]UK Centre for Ecology & Hydrology, Environment Centre Wales, Bangor, Wales, UK
[6]Environment & Geography Dept, University of York, York, England, UK

*Correspondence to*: Per Erik Karlsson (pererik.karlsson@ivl.se)

**Abstract.**

The annual, accumulated stomatal ozone uptake during the vegetation season, i.e. the species-specific Phytotoxic Ozone Dose above a threshold of 1.0 nmol $m^{-2}$ $s^{-1}$, $POD_1SPEC$, was estimated for European forest tree plant functional types for the years 2008 – 2012. These $POD_1SPEC$ estimates were based on ozone concentrations simulated with the EMEP CTM model in combination with stomatal ozone uptake estimated with the $DO_3SE$ (Deposition of Ozone for Stomatal Exchange) model. $POD_1SPEC$ -based dose-response relationships for impacts of ozone on forest growth rates were constructed based on results from multi-year experiments with young trees generated within the framework of the UNECE LRTAP Convention. Official information on forest gross growth rates as well as natural and harvest removals for different European countries for the years 2008 – 2012 were used to estimate annual changes in forest living biomass carbon (C) stocks under two different scenarios, with and without the negative impacts of ozone on forest gross growth rates, estimated using the $POD_1SPEC$ -based dose-response relationships for impacts. This resulted in estimates of the annual gap between forest gross growth and the total removals, i.e. the annual forest stock changes, under current negative ozone impacts, as well as in the absence of negative ozone impacts. Estimates were made by collating any species-specific information into broad European coniferous and deciduous forest types for consistency with forest statistics. The default IPCC methodology was used to convert estimates of the impacts of ozone on the annual changes in forest living biomass C stocks. The results showed that the critical level for negative impacts on forests suggested by the UNECE LRTAP Convention, based on $POD_1SPEC$, was exceeded in large parts of Europe during 2008 - 2012, except for inland areas in the Mediterranean and for small parts of Continental Europe as well as for and the Fennoscandian mountain range. The highest $POD_1SPEC$ was estimated for the coastal regions of mid-latitude Europe including the UK, limited to the north by mid-Sweden and south Norway and Finland. To the south, lower values for $POD_1SPEC$ were estimated for most of the Iberian Peninsula as well as parts of the Mediterranean coastal regions. It was



estimated that reduced ozone exposure, similar to pre-industrial conditions, would increase European forest stem volume growth rates by 9%, but it would increase European forest annual net changes in standing stocks by 28%. The difference in

gross forest stem volume growth with and without ozone impacts was relatively similar in for example Germany and France. However, since the gap between gross growth and total removals was much smaller for Germany, the enhanced growth in the absence of ozone had a much larger relative impact on the forest standing stock changes in Germany, compared to France. Summarized for all European forests, the C sequestration to the living biomass C stock was estimated to increase by 31% in the absence of ozone exposure. A thorough review of the literature resulted in the conclusion that mature trees under field

conditions cannot be assumed to be less sensitive to ozone exposure compared to young trees under experimental conditions strongly suggesting these results are credible for European forest stands of different age classes.

**Short summary**

Stomatal ozone uptake and the negative impacts on forest growth rates were estimated for European forests. This was translated

to annual increments in the forest living biomass carbon stocks, with and without ozone exposure. In the absence of $O_3$ exposure, European forest growth rates would on average increase by 9%, but the sequestration to the living biomass carbon stocks would increase by 31%, since the sequestration depends on the difference between growth and harvest rates.


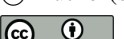



## 1 Introduction

Forests are important for cost-effective, land-based climate mitigation (Griscome et al., 2017; Roe et al., 2021), including carbon (C) sequestration by the increases in the forest ecosystem C stocks (Pan et al., 2011). Furthermore, long-lived biogenic raw materials produced by forests can store C in harvested wood products as well as substitute for the use of fossil based

materials (Sathre and Gustavsson, 2009; Jasinevičius et al., 2015; Gustavsson et al., 2017). The yearly net C sink in global forests ecosystems has been estimated at 4.000 Mt $CO_{2e}$ (Pan et al., 2011) of which approximately 80% is stored in the living biomass C stocks. The importance of European forests in this capacity has been clearly demonstrated (Hyyrynen et al., 2023; Korosuo et al., 2023). European forests are likely to absorb ~10 % of all European anthropogenic $CO_2$ emissions (Janssens et al., 2003). The C stocks in European forests above and below ground biomass are in the order of 48.000 M t $CO_{2e}$ and it has

been estimated to have increased annually over the time period 2010 – 2020 by 560 M t $CO_{2e}$, i.e. by 1.4% annually (Forest Europe, 2020).

Forest growth rates in many European countries are primarily determined by forest management (Etzold et al., 2020). In general, forests that are actively managed sequester C at higher rates than non-managed forests (Nabuurs et al., 2008). The C sequestration of managed forests to a large extent depends on the balance between forest growth and removals, including

harvests as well as natural losses (Soimakallio et al., 2021). Any measures that increase the productivity of temperate or boreal forest are likely to increase the forest C sequestration.

Forest growth rates are also affected by environmental conditions such as meteorological variables and air pollutants. Tropospheric ozone ($O_3$) is an air pollutant that has been found to cause substantial losses to tree biomass (Emberson, 2020). $O_3$ impacts on vegetation is estimated based on the accumulated amounts of $O_3$ that is taken up to the leaf interior through the

stomata during the vegetation season, i.e. the Phytotoxic Ozone Dose, POD (Mills et al. 2011). Experimental studies fumigating young trees with $O_3$ for up to 10 years have been conducted across Europe for different tree species (Karlsson et al., 2007; Wittig et al., 2009). These experimental studies have been used within the UNECE Convention on Long-Range Transboundary Air Pollution (LRTAP) to develop POD-based dose-response relationships that can be used to identify areas across Europe where losses in whole tree biomass may occur due to elevated $O_3$ concentrations (LRTAP, 2017). Critical levels

for forest trees were set to prevent an annual biomass loss under experimental conditions exceeding between 2 (coniferous forests) and 4% (deciduous forests), based on above ground or whole tree biomass. Maps of ozone uptake and exceedance of POD critical levels (Simpson et al., 2007, 2022, Franz et al., 2017) show where forests might be at risk of damage due to $O_3$ pollution. Identifying areas of such critical level exceedance can been used within LRTAP to apply an effects-based emission reduction policy (Emberson, 2020).


It is important to estimate actual $O_3$ impacts to forest trees under ambient $O_3$ pollution concentrations, since this would allow assessment of $O_3$ impacts on forest health (Marzuoli et al., 2019), productivity (Karlsson et al., 2005) and C sequestration rates, the latter becoming increasingly more important as nature-based solutions are promoted across Europe to increase



climate resilience (Calliari et al., 2022). Due to the multi-year lifetime of forest trees, $O_3$ impact studies involve a time component over which effect estimates are integrated. Many $O_3$ impact studies for trees report only the percent reduction of biomass caused by $O_3$ at the end of the experiment and information on the biomass at the start of the experimental fumigation with $O_3$ is many times not provided. As a result, impacts on tree growth rates cannot be calculated directly. The significance of this problem increases at low growth rates in relation to the size of the $O_3$ effect (Karlsson, 2012). Information about $O_3$ impacts on forest growth rates is necessary for relevant assessments of long-term impacts of $O_3$ on forest growth and C sequestration (Korosuo et al., 2023). The use of the whole tree biomass, POD-based, dose-response relationship can then be applied for assessments of long-term impacts of $O_3$ on forest C sequestration in combination with methods (such as forest growth models) that are able to simulate $O_3$ effects on growth rates in a complex forest environment (Sitch et al., 2007; Franz et al., 2018; Subramanian et al., 2015; Otu-Larbi et al. 2020).

The first phase of the Tropospheric Ozone Assessment Report (TOAR; https://igacproject.org/activities/TOAR/TOAR-I ) built the world's largest database of $O_3$ metrics to identify the global distribution of the pollutant and trends in $O_3$ concentrations over time. The second phase of TOAR (https://igacproject.org/activities/TOAR/TOAR-II ), to which this paper contributes, has a broader scope, with the aims being to investigate the impact of tropospheric $O_3$ on human health and vegetation. The present work will address these goals by assessing the effects of ozone on C sequestration in forest living biomass. Hence, in this study we re-analyse existing whole tree biomass, POD-based, dose-response relationships based on multi-year experiments with young trees to provide estimates of $O_3$ impacts on the tree growth rates and changes in the forest living biomass C stocks. Consequently, $O_3$ impacts on European forest gross growth rates can be estimated using the LRTAP experimental information (LRTAP, 2017) in combination with readily available forest statistic information describing annual increments in forest stocks (Forest Europe, 2020, UNECE, 2011).

The overall aim of this study was to apply LRTAP air quality guidelines to estimate the impacts of the present $O_3$ exposure, based on POD, on the C sequestration to the living biomass C stocks of Europe´s managed forests. This aim was achieved by addressing the following research questions:

1.      How does POD vary spatially for key coniferous and deciduous forest tree species and functional types across Europe?

2.      How can $O_3$ dose – response relationships for biomass reduction be converted into growth rate relationships for key European forest tree species and forest plant functional types?

3.      How can impacts of $O_3$ exposure on the C sequestration of living biomass C stocks of forests be estimated, based on the difference between forest gross growth rates and total removals, the latter including both harvests and natural losses?

4.      What is the magnitude of the negative impacts of present $O_3$ exposure on the C sequestration of the living biomass C stocks in forests, aggregated at both the European and national levels?



## 2 Methods

### 2.1 Overview of methods

In Figure 1 it is described how datasets and models are integrated and used to quantify the effect of $O_3$ on changes in annual
stem volume C stock increments (Mt $CO_{2e}$ yr$^{-1}$) of managed forests across Europe. Gridded surface meteorological data and
$O_3$ concentration data at ca. 50 m height (assumed to be the top of the atmospheric surface layer) for Europe were provided
from the EMEP MSC-W chemical transport model (CTM) (Simpson et al., 2007, 2012). Version rv4.35 of the EMEP model
(Simpson et al., 2020), as used here, is driven by 3-hourly meteorological data from the European Centre for Medium Range
Weather Forecasts Integrated Forecasting System (ECMWF-IFS, https://www.ecmwf.int/en/research/modelling-and-
prediction). These meteorological data are interpolated to hourly resolution inside the EMEP model, and variables such as
cloud-cover and vapour pressure deficits (VPD) calculated. Anthropogenic emissions data for the EMEP model are
predominantly derived from official national estimates (Mathews et al., 2020), and biogenic emissions (isoprene, terpenes, etc)
calculated as in Simpson et al. (2012). These calculations used 20 vertical layers, and $O_3$ data from the lowest vertical layer
(which approximates to an average height of 50 m above the surface) were used in this study. For consistency with the land
cover mapping data used in this project (Cinderby et al., 2007), all data were provided at a horizontal resolution of ~ 50 x 50
km$^2$ on the polar-stereographic which was used until 2016 by EMEP (https://www.emep.int/mscw/emep_grid.html).





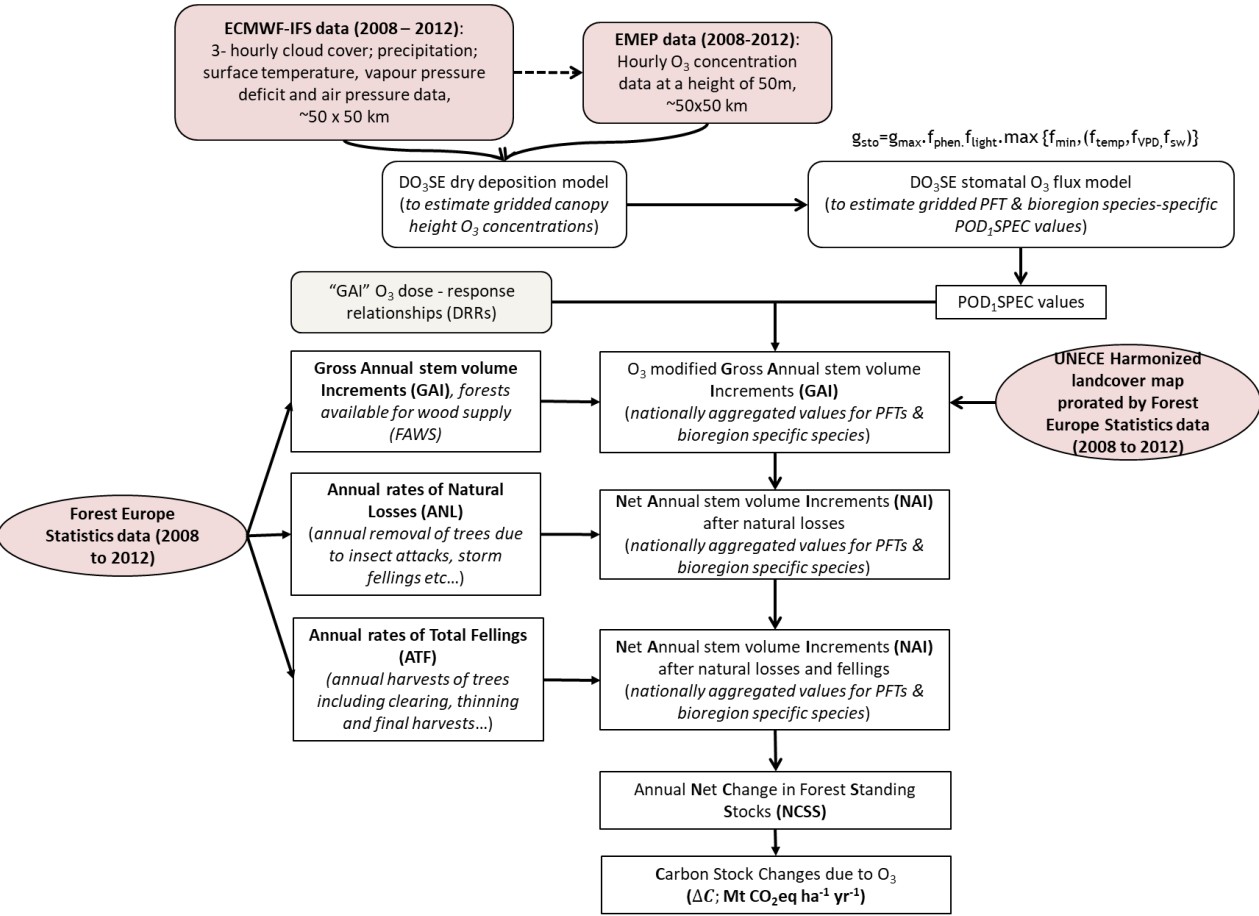

**Figure 1: Overview of the datasets and methods used to estimate the influence of O₃ on forest annual living biomass C stock**
**increments (Mt CO₂eq yr⁻¹) for the time period 2008 - 2012.**

The DO₃SE (Deposition of Ozone for Stomatal Exchange, Emberson et al., 2000; 2007; Büker et al., 2012; 2015) dry

deposition model estimates gridded O₃ concentrations down to the forest canopy height. From this, the stomatal O₃ flux values,

i.e. the POD₁SPEC (Phytotoxic Ozone Dose above a threshold of 1 nmol m⁻² Projected Leaf Area s⁻¹ for Specific tree species),

are calculated using the multiplicative stomatal conductance module, parameterised for bioregion-specific species (determined

according to the biogeographic regions of Europe (EEA, 2016) as well as plant functional types (PFTs). POD₁SPEC values

are used with gross annual stem volume increment (GAI) dose-response relationships (DRRs) to estimate the effect of O₃ on

GAI. These grid level estimates are aggregated to the national level using forest coverage data from the UNECE harmonized

land cover map (Cinderby et al., 2007) scaled according to the mean 2008 to 2012 estimates of the total forest cover (Forest

Europe, 2020, UNECE, 2011). This results in country level estimates of GAI, with and without O₃ pollution. Country level



GAI values are converted to net annual stem volume increment (NAI) values by allowing for forest removals by both natural causes and fellings. Country values for annual net change in forest standing stocks (NCSS), both with and without $O_3$ pollution, are converted to changes in total C stocks using default methodologies of the IPCC (Penman et al., 2003), which include both above and below ground living biomass. This allows the effect of $O_3$ on annual C sequestration in the living biomass of
managed European forests to be estimated.

### 2.2 Calculating $POD_1SPEC$

The $DO_3SE$ deposition model was used in an off-line mode to calculate total deposition of $O_3$. This allows the $O_3$ concentration at the top of the forest canopy (i.e. at the interface between the atmosphere and the canopy-influenced boundary layer) to be estimated for the different forest types across Europe. This off-line calculation uses the $DO_3SE$ stomatal flux model
parameterisations provided in Table S1 to estimate the stomatal and non-stomatal deposition sink for $O_3$ from which the canopy height $O_3$ concentration $c(z\_1)$ in (nmol m$^{-3}$) at height $z\_1$(m) can be calculated as described in Eq. 1.

$$c(z1) = c(zRef) \cdot [1 - Ra(zRef, z1) \cdot VgRef] \qquad [1]$$

where $c(zRef)$ is the concentration of $O_3$ from ca. 50m, assumed to represent the top of surface layer in the EMEP model, $Ra(zRef, z1)$ is the atmospheric resistance to $O_3$ transfer between zRef and z1, and VgRef is the deposition velocity of the tree canopy. It is important to note that this "big-leaf" calculation of $c(z1)$ makes use of big-leaf resistance and conductance terms, which differ from the leaf-level values used below (see Tuovinen et al., 2008, for more discussion of these distinctions).
We define stomatal $O_3$ flux model parameterisations by species, bioregions and PFTs since we know that stomatal $O_3$ uptake
by forest trees is dependent upon tree physiology which varies by species and bioregion, the latter often causing different physiology in the same species (LRTAP, 2017). We define nine bioregions across Europe using the biogeographic regions of Europe map (EEA, 2016; see Table S2) and apply the appropriate stomatal conductance (g_sto) model parameterisation (Table S1) in the estimation of g_sto following Eq. [2]; see the UNECE Mapping Manual (LRTAP, 2017) for further details.

$$g_{sto} = g_{max} \cdot f_{phen} \cdot f_{light} \cdot max\{f_{min}, (f_{temp} \cdot f_{VPD} \cdot f_{PAW}\} \qquad [2]$$

Where $g_{sto}$ is the actual stomatal conductance (mmol $O_3$ m$^{-2}$ PLA s$^{-1}$, PLA, projected laef area) and $g_{max}$ is the species-specific maximum $g_{sto}$ (mmol $O_3$ m$^{-2}$ PLA s$^{-1}$) and the parameters $f_{phen}$, $f_{light}$, $f_{temp}$, $f_{VPD}$, $f_{PAW}$ and $f_{min}$ are all expressed in relative terms. $f_{phen}$ allows for the variation in $g_{sto}$ during the growing season, $f_{min}$ defines a daytime minimum $g_{sto}$ and the remaining factors
represent the modifying influence of environmental variables (irradiance, temperature, atmospheric water vapour pressure deficit and plant available soil water respectively.





Stomatal conductance model parameterisations only exist for a limited number of forest tree species/ PFTs, namely: birch (*Betula pendula*), beech (*Fagus sylvatica*), Norway spruce (*Picea abies*), Scots pine (*Pinus sylvatica*), Aleppo pine (*Pinus halepensis*), temperate deciduous oak (*Quercus robur*), Mediterranean deciduous oak (*Quercus faginea*, *Quercus pyrenaica* and *Quercus robur*) and Mediterranean evergreen oak (Holm oak, *Quercus ilex*). Table S2 describes the combination of stomatal $O_3$ flux parameterisations (to estimate the $POD_1SPEC$ values) and GAI DRRs (to estimate the damage caused by $POD_1SPEC$ that are used in each bioregion along with the number of EMEP grids these bioregions cover). Table S2 shows that for those bioregions that comprise > 100 EMEP grids, between 29 and 85% of forest area is represented by these specific species, the remainder being classified as other deciduous, other coniferous or mixed and represented by an appropriate species of that bioregion and forest type.

The stomatal $O_3$ flux ($F_{st}$) of sunlit leaves or needles of the upper canopy can then be calculated for forests across Europe using Eq. [3].

$$F_{st} = c(z_1) \cdot g_{sto} \cdot \frac{r_c}{r_b+r_c} \qquad [3]$$

where $g_{sto}$ is in m s$^{-1}$, $r_b$ is the leaf quasi-laminar resistance and $r_c$ the leaf surface resistance, both given in s m$^{-1}$. For further details on the resistance scheme, see UNECE Mapping Manual (LRTAP, 2017).

The accumulated species-specific $POD_1SPEC$ (mmol $O_3$ m$^2$ PLA) was calculated according to Eq. [4] by summing modelled hourly $F_{st}$ values ($F_{sti}$) over the smaller of either a 6-month period (1 April to 30 Sept) or a species-specific period defined by the forest latitude model (LRTAP, 2017).

$$POD_1SPEC = \sum_{i=1}^n \left[F_{st_i} - y\right] \text{ for } F_{st_i} \geq y \text{ nmol m}^{-2} \text{ projected leaf area (PLA) s}^{-1} \qquad [4]$$

Where $F_{sti}$ is the hourly mean $O_3$ flux in nmol $O_3$ m$^{-2}$ PLA s$^{-1}$, and n is the number of hours within the accumulation period. The threshold of 1 nmol $O_3$ m$^{-2}$ PLA s$^{-1}$ (Mills et al., 2011; LRTAP, 2017) represents the ability of plants to detoxify a certain amount of $O_3$. $POD_1SPEC$ was calculated separately for coniferous and deciduous tree species and separately for scenarios with (f_PAW) and without (NSW) the influence of the estimated soil water deficit (Bueker et al., 2012).

**2.3 Deriving $POD_1SPEC$ dose-response relationships for gross annual stem volume increment**

$POD_1SPEC$ DRRs exist for a number of species/species groups and have been previously defined in Büker et al. (2015). The response parameter for these DRRs is a percentage reduction in total living biomass at the end of the experimental period,



corrected for the number of experimental years. As described in the introduction, this is not a suitable metric for use to estimate impact on long-term forest growth rates. To overcome this issue, we re-analysed the existing DRRs, converting the response variable from percentage reduction in total biomass to percentage reduction in GAI. This was achieved by defining a standard sigmoidal growth function for tree biomass based on the Richards equation [Eq. 5] (Richards, 1959).

$Tree\ biomass = y_0\left[1 - exp^{(-y_1 \cdot\ tree\ age)}\right]^{\left(\frac{1}{1-y_2}\right)}$                    [5]

Where $y_0$, $y_1$ and $y_2$ are set to 1, 0.03 and 0.65 based on data collected for key European tree species from sites across Europe (Fellner & Rechberger, 2009; shown in Figure S1). This allows calculation of the tree biomass at the beginning of each $O_3$ fumigation period (which usually starts when the tree is a few years old) and the GAI that would be expected each year until

the end of the fumigation period. For each experiment, the $O_3$ effect is estimated as the average percentage reduction on annual growth rate of each $O_3$ treatment vs control, assuming that the relative $O_3$ effect on growth rate is constant, irrespective of age of tree or GAI.

The GAI DRR is estimated by combining results of fumigation experiments conducted on a particular species or species group

following the method of Fuhrer (1994). This estimates a regression for each individual experiment to define the biomass at zero $O_3$ exposure. This is then used to scale the different treatment effects so that zero exposure is always associated with no effect at the individual experiment level. The percentage reductions in GAI for each experiment and treatment are pooled and a linear regression is drawn through the data to give a species- or PFT-specific DRR.

The UNECE harmonised land cover map (Cinderby et al., 2007), which describes coverage of individual forest tree species at a spatial scale of 1 x 1 km are aggregated to the ~50 x 50 km EMEP grids to provide an area-weighting for each species and PFT across the EMEP modelling domain. This area weighting is scaled according to the national forest area coverage data from the Forest Europe statistics (Forest Europe, 2020, UNECE, 2011) to ensure that national total forest cover is consistent with the GAI data.

**2.4 Estimating the influence of $O_3$ on gross annual stem volume increments.**

The species/ PFT specific DRRs are used to estimate the GAI under per-industrial $O_3$ concentrations (assumed constant at 10 ppb across the whole of Europe, after Volz & Kley, 1988) for each species/PFT in each grid. This allows an estimate of the reduction in GAI due to the occurrence of $O_3$ pollution for each grid, which is then scaled to provide a country area weighted value for coniferous and deciduous (broadleaf) PFTs scaled, according to land cover.




**2.5 Estimates of O₃ impacts on European forest carbon sequestration**

Forest statistics for European countries are available at the national level from the Forest Europe statistics (https://fra-data.fao.org/assessments/panEuropean/2020/FE/home/overview , section "Increment and fellings, in forest available for wood supply", indicator 3.1.). At the commencement of this study, information was available for forests at the national level for five-
year periods until the period 2010, which is based on mean values for the period 2008 - 2012. In most cases information is presented separately for coniferous and broadleaf tree species. Hence, calculations of O₃ impact on European forest C sequestration were made separately for each country and separately for coniferous and broadleaf tree species. The assessment thus regards the annual values for the time period 2008-2012. Calculations were restricted for forests available for wood supply (FAWS). For most countries, the ratio of the area of total forest land to the area of FAWS did not exceed 1.3 (data now shown).
Annual net changes in forest standing stocks, i.e. the summed stem volumes of living trees, were calculated according to the scheme outlined in Fig 2.

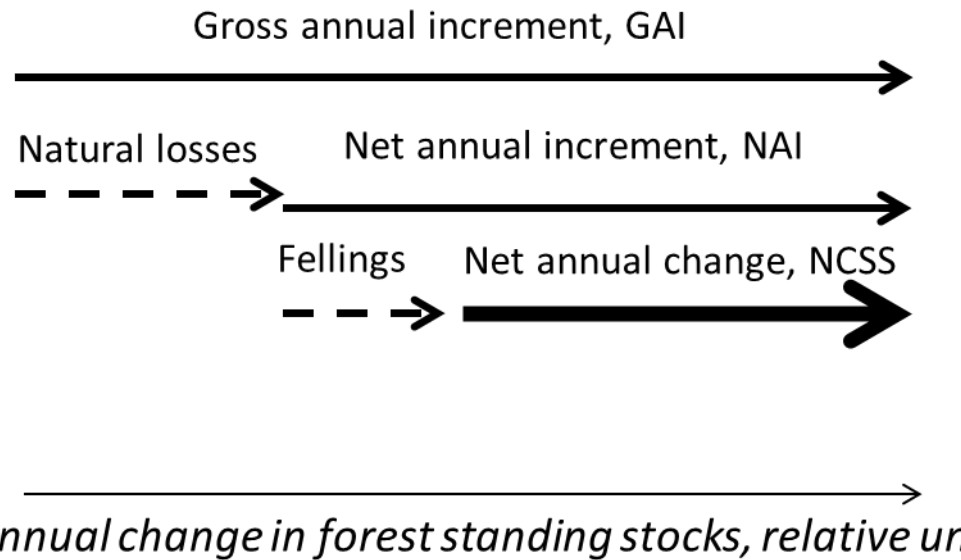

**Figure 2. An illustration of the scheme used to calculate the net changes in the forest standing stocks. Modified from Forest Europe**
**(2020) and Korosuo et al. (2023). The arrow with the thickest line represents the final value calculated as the net forest standing stock change (NCSS).**

The dependent variable for O₃ impacts on net changes of forest C stocks was the GAI in FAWS. The annual rates of natural losses (ANL) are subtracted from the GAI to give the Net Annual stem volume Increment (NAI). It was assumed that O₃
exposure does not have an impact on the ANL, mainly since there are, as far as we are aware, no published studies on O₃ impacts on tree mortality. NAI is then subtracted by the annual rates of total fellings (ATF), including fellings of both living



and dead trees as well as including harvest residues, i.e. all biomass that is removed from the FAWS. The resulting values will be the annual net change in forest standing stocks (NCSS) in FAWS. Information was not available for all parameters listed above for all countries. Missing values were replaced based on the relations between the relevant parameters for nearby

countries positioned within the same European forest region. The regions used were those used by Forest Europe (2020), i.e. North Europe, Central-West Europe, Central-East Europe, Southwest Europe, Southeast Europe and finally Russia with neighbouring countries.

The forest statistics for different European countries only provide information on the GAI, natural- and harvest removals of

the total forests together with information for the different areas of coniferous and broadleaf forests. Hence, the share of the GAI, natural- and harvest removals between coniferous and broadleaf forests has to be assumed based on the differences in the forest areas. This assumption was tested and found to be relatively correct for Sweden, a country for which the relevant forest statistics were available separately for coniferous and broadleaf forests (data not shown).

The NCSS was converted to C stock changes as described in IPCC's "Good Practice Guidance for Land Use, Land-Use Change and Forestry" (Penman et al., 2003).

$$\Delta C = I_v * BEF * (1+R) * D * CF$$

Where $\Delta C$, C sequestration to tree living biomass (tonnes C ha$^{-1}$ yr$^{-1}$); $I_v$, yearly increment of timber volume (m$^3$ ha$^{-1}$ yr$^{-1}$); D, density stem (tonnes dry weight m$^{-3}$); CF, "carbon fraction", of dry matter (tonnes tonnes$^{-1}$); BEF, biomass expansion factor, converts between stem biomass and total living biomass above ground; R, shoot/root ratio. The value of $\Delta C$ was then converted to $CO_2$-eqvivalents ($CO_{2e}$) by multiplying with 3.67. The constants used in the present study are shown in Table 1.

**Table 1. Constants used for the conversion from change in forest standing stock to changes in forest C stock, based on IPCC (Penman et al., 2003, Ågren et al., 2021).**

|  | Coniferous tree species[*] | Broadleaf tree species[*] |
|---|---|---|
| D, t dry mass m$^{-3}$ | 0.41[**] | 0.55[***] |
| BEF, biomass above ground expansion factor (dimensionless) | 1.125[#] | 1.15[##] |
| R, root/shoot ratio | 0.32 | 0.26 |
| CF, carbon fraction of dry matter (t t$^{-1}$). | 0.51[###] | 0.47[###] |





[*] Valid for forest stands with standing biomass of 50-150 t ha[-1] for coniferous and 75-150 t ha[-1] for broadleaf species; [**] mean value for *Picea abies* and *Pinus sylvestris*; [***] mean value for *Betula* sp. and *Fagus sylvatica*; [#] mean value for *Picea abies* and *Pinus sylvestris*, boreal and temperate forests; [##] mean for broadleaf forests, boreal and temperate; [###] Ågren et al., 2021;


## 3 Results

The spatial distribution of the estimated $POD_1SPEC$ (mmol $O_3$ m$^{-2}$ PLA) across Europe is shown in Fig. 3, separately for coniferous and deciduous tree species, as an average value for the years 2008 to 2012. To give an indication of the variation in $POD_1SPEC$ between years, the differences between the minimum and maximum $POD_1SPEC$ values across 2008 to 2012

for conifers and deciduous species are also shown.





(a)

(b)

(c)

(d)



(e)                                                    (f)

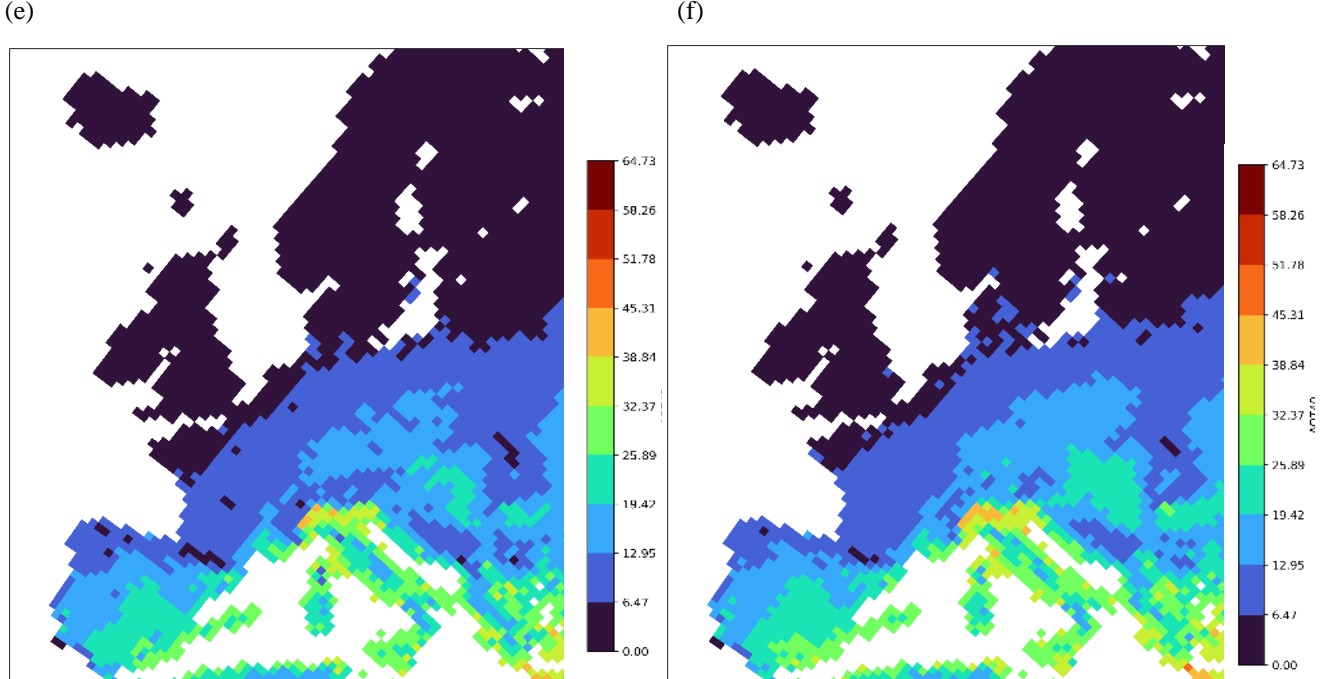


**Figure 3. The spatial distribution of the estimated POD₁SPEC (mmol O₃ m⁻² PLA) across Europe accumulated during the growing season using the f_PAW scenario, separately for coniferous (a) and broadleaf (c) tree species for annual mean values for the period 2008-2012 and as the difference in maximum and minimum POD₁SPEC values across the years 2008-2012 (coniferous, b; broadleaf, d). Also shown is the mean annual value of AOT40 in ppb.hrs for the period 2008-2012 and the Continental Central Europe (CCE) growing season of Norway Spruce (e) and Beech (f).**

The highest POD$_1$SPEC values were estimated for the coastal regions of mid-latitude Europe including UK, for both coniferous (15 to 20 mmol O$_3$ (m$^2$ PLA)$^{-1}$) and broadleaf (22 to 28 mmol O$_3$ (m$^2$ PLA)$^{-1}$) forests, limited to the north by mid-Sweden and

south Norway and Finland. There were even high values estimated for the southern part of Iceland. To the south, lower values for POD$_1$SPEC were estimated for most of the Iberian Peninsula as well as all the Mediterranean coastal regions. There were high values estimated for POD$_1$SPEC also in parts of the alpine region and some surrounding regions of inland continental Europe (Fig. 3a, c). There were low values (5 to 17 mmol O$_3$ (m$^2$ PLA)$^{-1}$) for both coniferous and broadleaf deciduous estimated in particular for the inland of the Iberian Peninsula but also for the eastern parts of continental Europe as well as for the Balkan

region. For coniferous forests, there were low values (around 10 mmol O$_3$ (m$^2$ PLA)$^{-1}$) estimated for POD$_1$SPEC also for parts of Italy. In general, POD$_1$SPEC values were considerable higher (by almost 10 mmol O$_3$ (m$^2$ PLA)$^{-1}$) for deciduous as compared to coniferous trees (Fig. 3a, c). The inter annual differences (Fig. 3b, d) are also greater for broadleaf deciduous





(often reaching up to 10 mmol $O_3$ ($m^2$ PLA)$^{-1}$) compared to coniferous with inter annual variability over the five years of only up to around 5 mmol $O_3$ ($m^2$ PLA)$^{-1}$. These differences are not overtly driven by soil water stress as might be expected since

the NSW model runs also show a similar magnitude of variability, albeit over a slightly reduce extent across Europe (see Figure S2). Finally, it is clear from Fig. 3a, e and f that the AOT40 metric has a very different spatial pattern with a strong north to south gradient, identifying highest values (over 30 ppm.hrs) and hence risk in Mediterranean and mid to southern continental Europe.

Figure 4 gives an indication of which of growing season or key environmental variables (i.e. which of $f_{temp}$, $f_{VPD}$, $f_{PAW}$) are most limiting of POD$_1$SPEC across Europe. The length of the growing season may cause a limitation in the Northern latitudes when growing season length is below around 135 days (Figure 4a), however it is likely that temperature would also cause a limitation outside of the growing season. This is clear from the function used to describe the temperature limitation of stomatal conductance (Figure 4b) showed homogenously high values across most of Europe, except for the alpine region as well as for

northern Fennoscandia, Scotland and Iceland. The function used to describe the VPD limitation of stomatal conductance (Fig. 4c) showed a geographical variation that more resembled the pattern for the estimated POD$_1$SPEC, with higher values for continental coastal regions and lower values for continental inland regions. The function used to describe the soil water deficit limitation of stomatal conductance, $f_{PAW}$, (Fig. 4d) showed low values only for the inland Iberian Peninsula.






(a)  Growing season length (days)          (b) $f_{temp}$

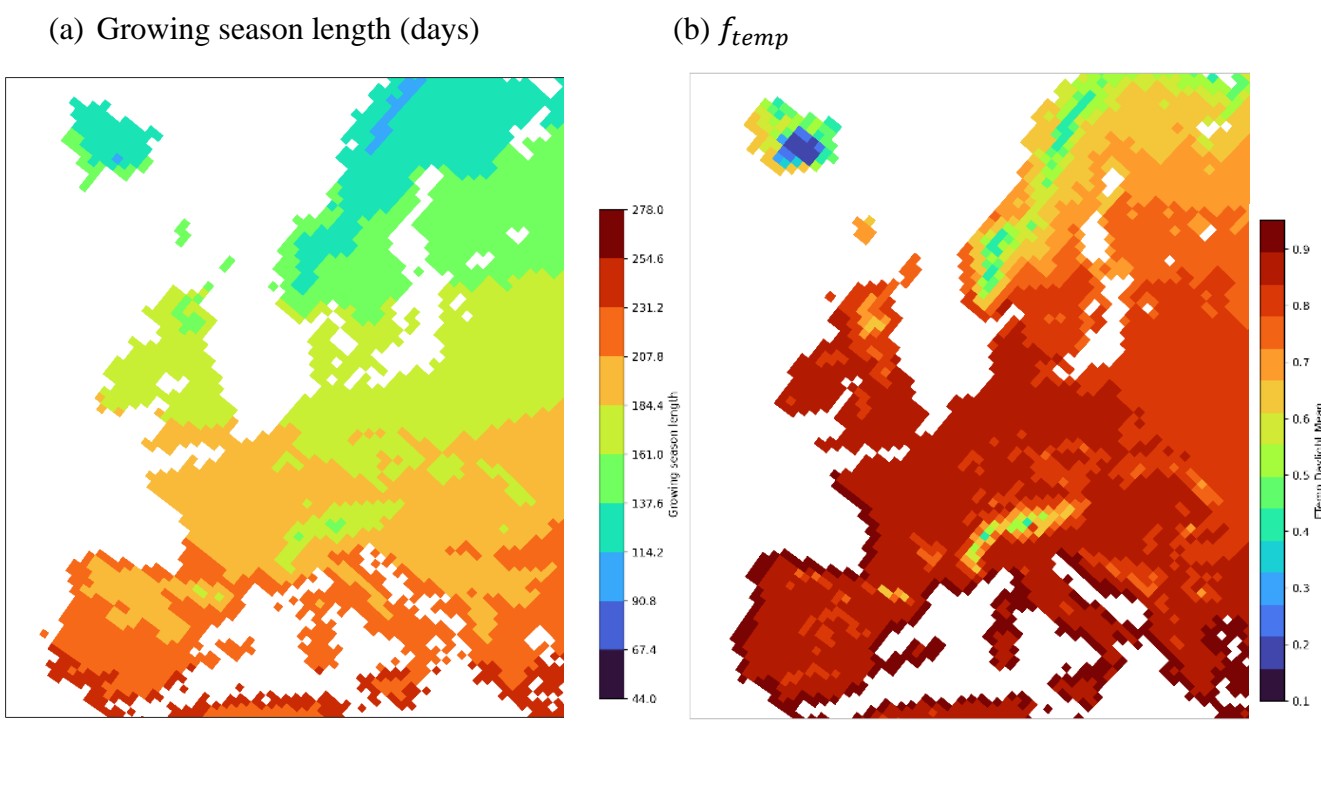

(c)        $f_{VPD}$                          (d) $f_{PAW}$



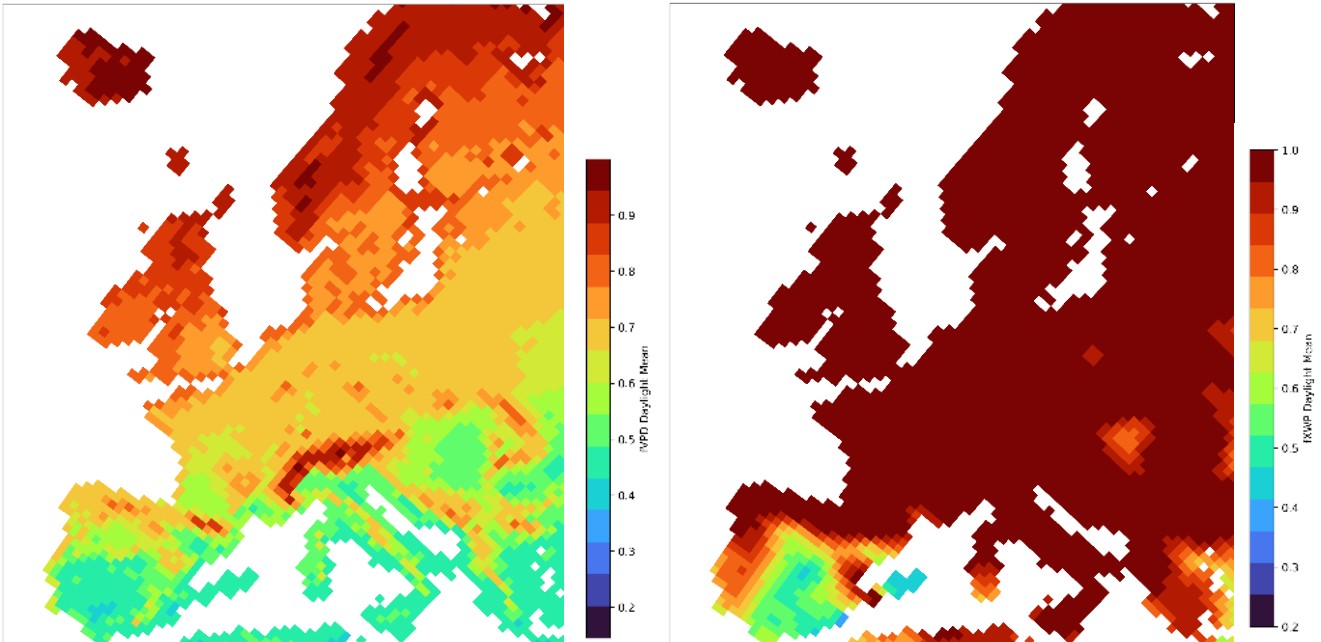

**Figure 4. The limits to POD$_1$SPEC from each of the key three environmental variables (a) growing season length (in days) for broadleaf deciduous forests, (b) f$_{temp}$, (c) f$_{VPD}$ and (d) f$_{PAW}$) averaged during the growing season for all species and all years (2008-2012) across Europe. Grid values are mean values across all plant functional types for the growing season and for daylight hours only. Plots for individual species are given in the supplementary (Figure S4).**

The GAI DRRs developed from the re-analysis of the experimental fumigation data are shown in Fig. 5b for coniferous species and 5d for broadleaf deciduous species. The percentage reduction in whole tree living biomass (taken from Büker et al. (2015)) are also shown (Fig. 5a for coniferous and 5c for broadleaf deciduous) for comparison. The GAI DRR regressions are given in Table S3. The GAI DRRs show small improvements in the statistical relationship of the regression between relative GAI and POD$_1$SPEC, with $R^2$ values of 0.69 and 0.47 for coniferous and broadleaf deciduous respectively. The difference in the intercept at POD$_1$SPEC is negligible between relative biomass and relative GAI but the slope for the negative impacts was slightly more negative for the DRRs based on GAI growth rate reductions. These results suggest the relative GAI DRRs are as robust as the relative biomass DRRs and hence are suitable for use in risk assessment.





(a)

(b)

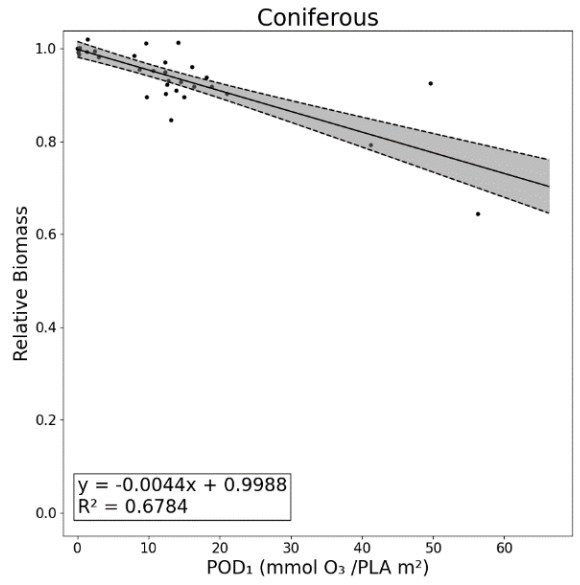

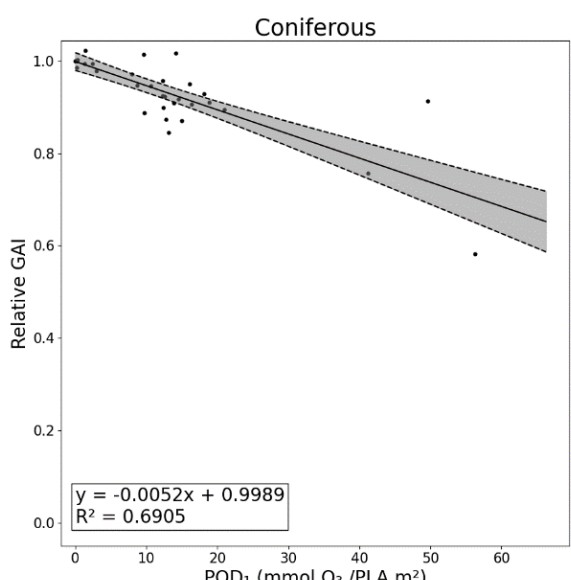

(c)

(d)

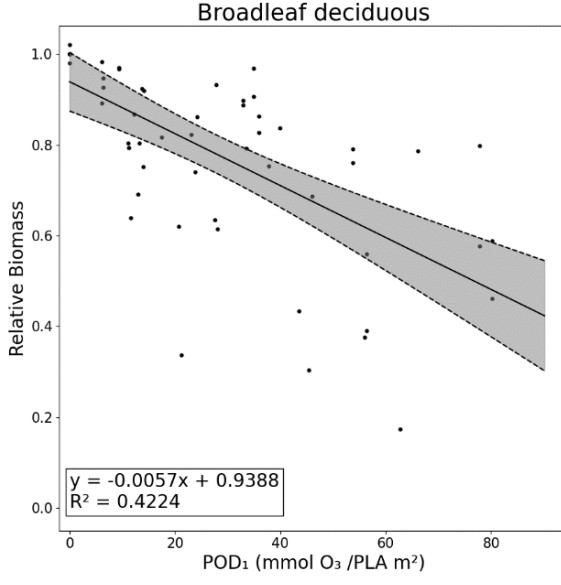

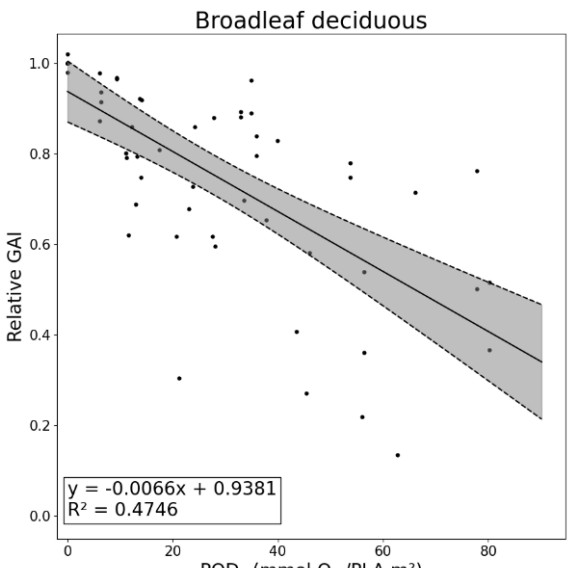




**Figure 5. Dose-response relationships (DRRs) based on POD₁SPEC for assessments based on impacts on the percent annual biomass loss (Relative biomass) (a, c) and the estimated impacts on gross annual increment (relative GAI) (b, d) separately for coniferous (a, b) and deciduous tree species (c, d). Other species and PFT dose-response relationships are given in the supplementary text (Table S3 and Figure S3).**


The estimated values for percent change in GAI, caused by the present $O_3$ compared to the absence of $O_3$ exposure, are shown in Table 2, calculated as mean values for the period 2008-2012 and mean values for different European countries. The reductions of GAI were calculated separately for coniferous and broadleaf deciduous tree species and separately for scenarios with (fPAW) and without (NSW) soil water deficits. The differences in reductions by $O_3$ exposure estimated by the two

different $O_3$ scenario, NSW and f$_{PAW}$, were very small (Table 2). Hence, only results based on the scenario f$_{PAW}$ will be used in the following.

The estimated reductions in growth rates by $O_3$ exposure were substantial. The largest growth reductions, estimated by including f$_{PAW}$, were found for deciduous tree species in Ireland, -32%, Poland (-28%) and Austria, (-28%) (Table 2). The

mean reductions in growth rates by $O_3$ for deciduous tree species across all countries was -17%. For coniferous tree species, the most negative values were estimated for Greece (-23%), Portugal (-19%) and Italy, (-17%). The mean reductions in growth rates by $O_3$ for coniferous tree species was -7%.



**Table 2. Percent reductions in gross annual increments (GAI), caused by the present O₃ exposure, calculated as mean values for POD₁SPEC for the period 2008-2012 and mean values for the area of different European countries. The estimated reduction of GAI was estimated based on O₃ dose-response relationships for impacts on growth rates (GAI DRRs) based on country-wide mean values for POD₁SPEC calculated separately for coniferous and deciduous tree species and separately for scenarios with (fPAW) and without (NSW) the functions for soil water deficit included.**

| | Coniferous, NSW | Deciduous, NSW | Coniferous, fPAW | Deciduous, fPAW |
|---|---|---|---|---|
| **Sweden** | -4 | -23 | -4 | -23 |
| **Norway** | -2 | -17 | -2 | -17 |
| **Finland** | -4 | -13 | -4 | -13 |
| **Denmark** | -5 | -15 | -5 | -15 |
| **Estonia** | -5 | -17 | -5 | -17 |
| **Latvia** | -2 | -24 | -2 | -24 |
| **Lithuania** | -4 | -14 | -4 | -14 |
| **Germany** | -4 | -20 | -4 | -20 |
| **Poland** | -1 | -29 | -0.5 | -28 |
| **Netherlands** | -4 | -18 | -4 | -18 |
| **Belgium** | -6 | -14 | -6 | -13 |
| **United Kingdom** | -4 | -23 | -4 | -23 |
| **France** | -12 | -8 | -12 | -8 |
| **Switzerland** | -1 | -26 | -1 | -26 |
| **Austria** | -4 | -28 | -4 | -28 |
| **Czech Republic** | -4 | -26 | -4 | -26 |
| **Hungary** | -10 | -5 | -10 | -4 |
| **Slovakia** | -5 | -14 | -5 | -14 |
| **Ireland** | -6 | -32 | -6 | -32 |
| **Slovenia** | -8 | -18 | -8 | -18 |
| **Bulgaria** | -7 | -17 | -7 | -17 |
| **Romania** | -10 | -8 | -9 | -8 |
| **Greece** | -26 | -13 | -23 | -11 |
| **Italy** | -17 | -18 | -17 | -18 |
| **Spain** | -13 | -15 | -9 | -10 |
| **Portugal** | -22 | -13 | -19 | -9 |






The estimated total, gross annual stem volume stock changes for all forests in Europe are shown in Fig. 6, with and without the exposure to present $O_3$ doses, calculated as $POD_1SPEC$. Also shown are the estimated total removals of forest stem volumes, including both natural as well as harvest removals. The rates of removals are assumed to be independent of $O_3$ regimes. Furthermore, in Fig. 6 the differences between the gross annual stock increments and the removals are indicated,

separately for the presence of $O_3$ impacts (continuous line arrow) and in the absence of $O_3$ impacts (dashed line arrow).

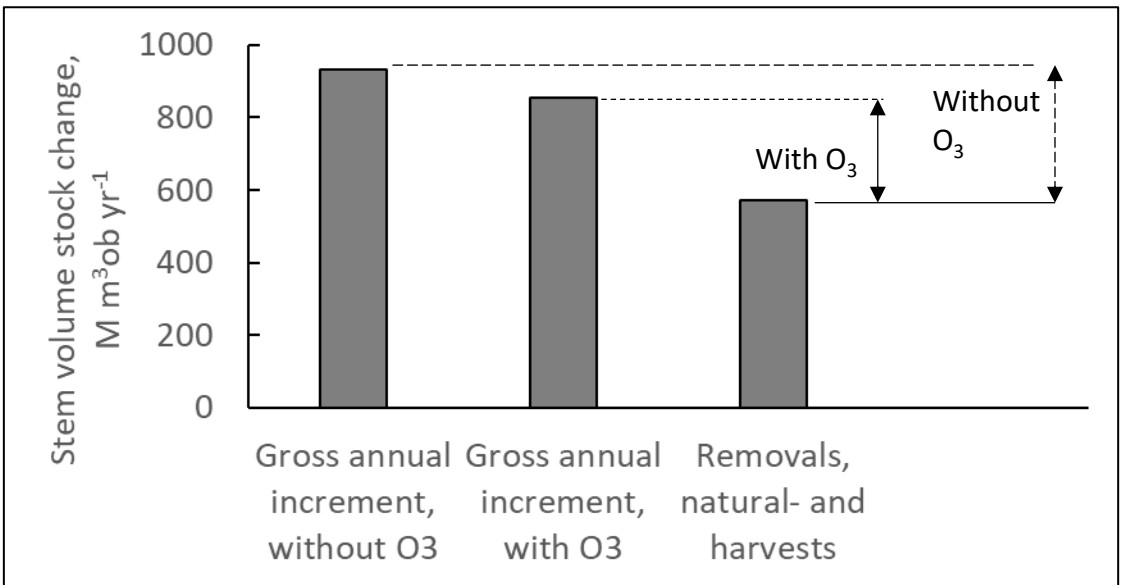

**Figure 6. The rates of total, annual gross stem volume increment growth for forests in Europe as mean annual values for the time period 2008-2012, including the sum of coniferous and deciduous tree species, with and without the exposure to present $O_3$ doses,**
**calculated as $POD_1SPEC$ with the scenario fPAW. Also shown are the estimated total removals of forest stem volumes, including both natural- as well as harvest removals. The rates of removals are assumed to be independent of $O_3$ regimes. Furthermore, the differences between the gross annual stock increments and the removals are indicated with vertical arrows, separately for the presence of $O_3$ impacts (continuous line arrow) and without the presence of $O_3$ impacts (dashed line arrow).**

The rates of annual, gross stem volume increment of all European forests in the presence of $O_3$ were 854 M $m^3$ over bark (o.b.) $yr^{-1}$ and in the absence of $O_3$ 933 M $m^3$ o.b. $yr^{-1}$ (Fig. 6). This is an increase of 9% in the absence of $O_3$. However, the annual net changes in forest standing stocks, NCSS, were 283 M $m^3$ o.b. $yr^{-1}$ in the presence of $O_3$ (continuous line arrow in Fig. 6) and in the absence of $O_3$, 363 M m3 o.b. $yr^{-1}$ (dashed line arrow in Fig. 6). This is an increase by 28% in the absence of $O_3$. Hence, the removal of $O_3$ exposure would increase European forest stem volume growth rates by 9%, but it will increase

European forest annual net changes in standing stocks by 28%. This illustrates the importance of not only considering $O_3$ impacts on forest gross growth rates but rather to consider the impacts on the gap between gross growth and the removals, i.e. net changes in forest standing stocks.



The annual gross stem volume increment for forests in the different European countries are shown in Fig. 7, with and without

the exposure to present $O_3$ doses. Also shown are the natural and harvest removals, which are assumed to be independent of the scenarios for $O_3$ exposure. The gap between the current rates of gross growth and the total removals differs considerably in magnitude between different countries. For instance, Germany has a relatively small gap between GAI and the total (natural + harvest) removals, while France has a much larger gap. The difference in GAI with and without $O_3$ exposure is relatively similar between these countries. However, this $O_3$ difference in GAI will have a much larger impact on the percent change in

NCSS for Germany, compared to France, since the gap between current GAI and removals is smaller.

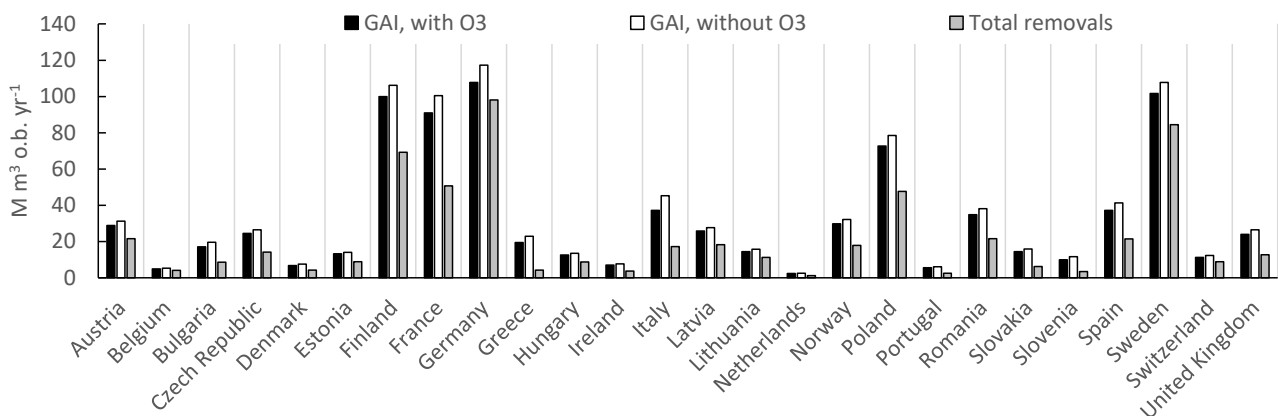

**Figure 7. The rates of annual gross stem volume increment growth for forests in different European countries, as annual values for the time period 2008-2012, including the sum of coniferous and deciduous tree species, with and without the exposure to present O3**

**doses, calculated as POD₁SPEC with the scenario fPAW. Also shown are the sum of the natural and harvest removals of forest stem volumes, i.e. the total removals. The rates of natural and harvest removals are assumed to be independent of ozone regimes. The countries are sorted left to right in alphabetical order.**

The change in the annual C sequestration in the forest living biomass C stock, in the absence compared to the presence of $O_3$

exposure is presented in Fig. 8. Results are presented for the individual Europeans countries, as absolute values (Fig. 8a) and differences without and with $O_3$ (Fig. 8b), separately for coniferous and deciduous as well as for total forests. The absence of $O_3$ exposure would increase the C sequestration to the living biomass, in absolute values, the most in Germany, France, Italy and Poland. Summarized for all European forests, the C sequestration to the living biomass C stock was estimated to be 343 M t $CO_{2e}$ $yr^{-1}$ in the presence of $O_3$ and 449 M t $CO_{2e}$ $yr^{-1}$ in the absence of $O_3$, i.e. an increase of 31%.




a

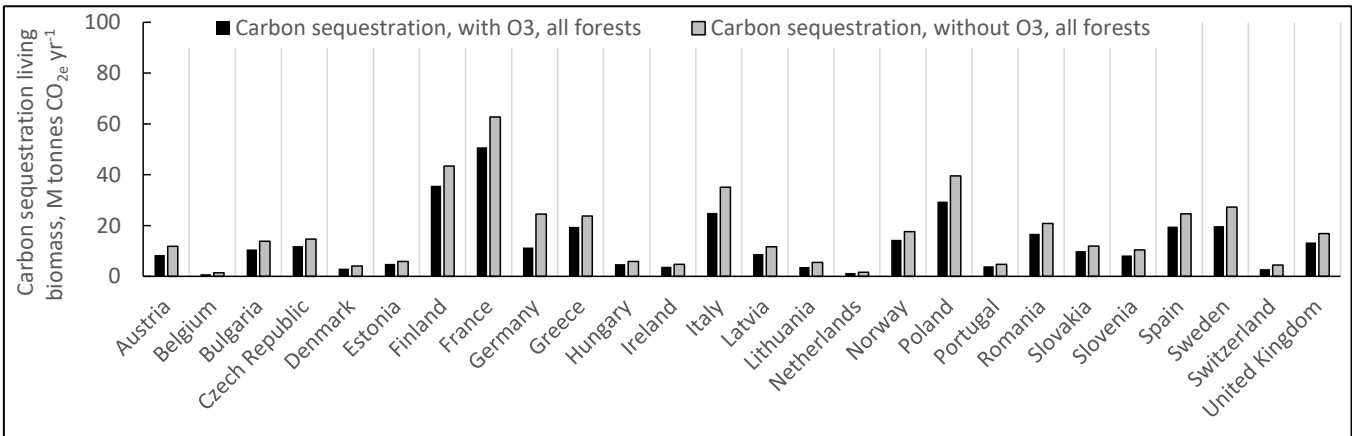

b

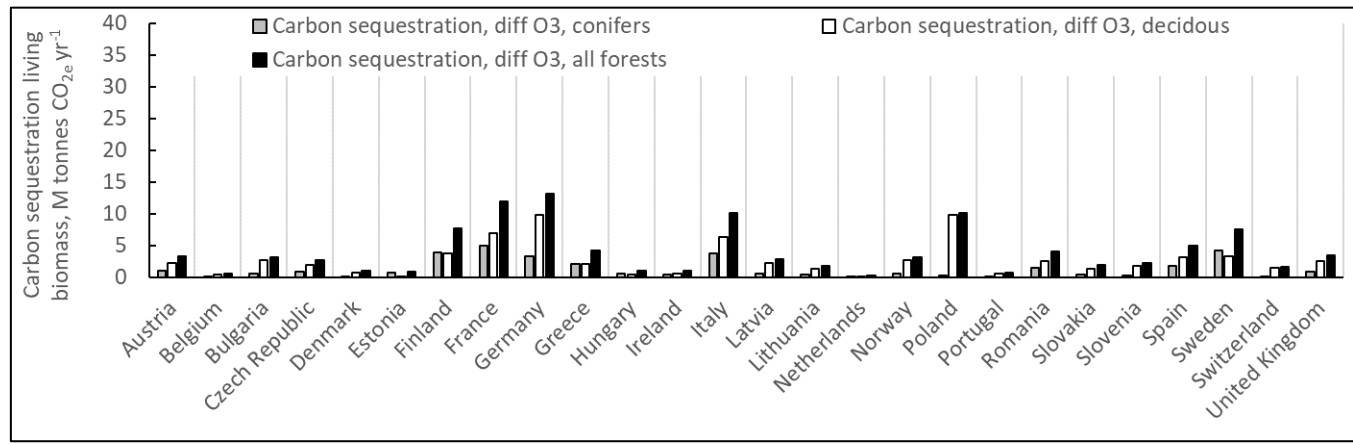


**Figure 8. The changes in the annual C sequestration in the living biomass C stocks, with and without the exposure to present O₃ doses, calculated as POD₁SPEC with the scenario fPAW. (a). absolute values for total forests; (b). differences without versus with ozone, separately for coniferous, deciduous and total forests. Results are presented for the individual Europeans countries, separately for coniferous and deciduous as well as for total forests. The countries are sorted left to right in alphabetical order.**


The percent changes in the annual C sequestration of forests in the different European countries, without the present O₃ exposure compared to with O₃ exposure, are shown in Fig. 9. Changes are shown separately for coniferous and deciduous forests as well as for total forests. As an example, the percent increase in the annual C sequestration of forests in the absence of O₃ is much larger for Germany, compared to France. This is because the annual C sequestration of forests in Germany in

the presence of O₃ was so low that the increase due to the removal of the O₃ exposure will be relatively large.



As mentioned already above, the percent increase in the C sequestration to the forest living biomass C stock of all European forests in the absence of $O_3$ is 31%. In general, the impact of the absence of $O_3$ is larger for deciduous compared to coniferous forests. This is because in the $O_3$ GAI DRRs, deciduous forests are considered more sensitive to $O_3$ impacts compared to

conifers. However, the estimates of the GAI of different tree functional types are somewhat uncertain since the forest statistics for different European countries only provides information on the GAI, natural- and harvest removals of the total forests together with information for the different areas of coniferous and deciduous forests.

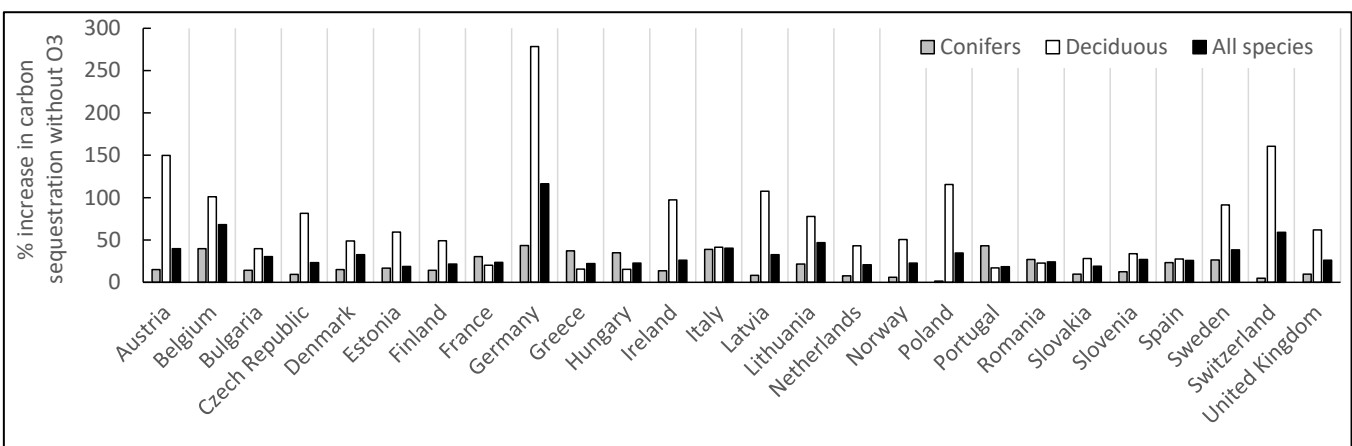

**Figure 9. The percent changes in the annual C sequestration of forests in the different European countries, without the present $O_3$ exposure compared to with $O_3$ exposure. Changes are shown separately for coniferous and deciduous forests as well as for total forests.**

## 4 Discussion

### 4.1 Spatial variation in POD$_1$SPEC

The UNECE Mapping Manual (LRTAP, 2017) sets the critical levels for $O_3$ impacts on trees in Europe, based on POD$_1$SPEC, to 9.2 mmol m$^{-2}$ for Norway spruce, related to a 2 % annual biomass reduction, and to 5.2 mmol m$^{-2}$ for Beech and Birch, related to a 4 % annual biomass reduction. The results shown in this study (Fig. 3a, c) suggest that this level was exceeded in large parts of Europe during the period 2008 - 2012, except for dry areas in the Mediterranean and for small parts of Continental Europe as well as for the Fennoscandian mountain range. This geographical distribution is very different from the exceedance

of the critical levels based on the ozone exposure index AOT40 applied by the European Union (10 ppm.h), which shows a strong latitudinal gradient, with the highest values in the Mediterranean (Figure 3e and f).

The estimates of the POD$_1$SPEC are the result of the combination of estimated $O_3$ concentrations and meteorological factors (f-functions) describing the limitation of the stomatal conductance (Emberson et al., 2000). The maps for the f-functions



provide some explanations for the patterns for $POD_1SPEC$ across Europe (Fig. 4). The function used to describe the temperature limitation of stomatal conductance (Figure 4b) showed in general homogenously high values across most of Europe. The function used to describe the VPD limitation of stomatal conductance (Fig. 4c) showed a geographical variation that more resembled the pattern for the estimated $POD_1SPEC$, with higher values for continental coastal regions and lower values for continental inland regions. Hence, this indicates that $f_{VPD}$ was an important factor for the estimates of $POD_1SPEC$

across Europe. The function used to describe the soil water deficit limitation of stomatal conductance, $f_{PAW}$, (Figure 4d) showed low values only for the inland Iberian Peninsula, explaining the low values for $POD_1SPEC$ estimated for this region. The reduced geographical extent of the $f_{PAW}$ limitation is likely due to the use of a rather insensitive $f_{PAW}$ relationship (with limitation starting only once soil water had been reduced to a level where only 25% remains available to the plant). This rather insensitive response was selected to ensure the protective effect of soil water stress in limiting $POD_1SPEC$ was not over

estimated. Future studies could explore the effect of species-specific $f_{PAW}$ in more detail especially since this was found to be an important variable determining $POD_1SPEC$ and crown defoliation and visible foliar injury a specific study sites in France, Italy and Romania (Sicard et al., 2020). To give an indication of the effect of a more sensitive $f_{PAW}$, Figure S4d shows $f_{PAW}$ values for each species for the year 2012 using an $f_{PAW}$ relationship that reduces $g_{sto}$ once 50% of available soil water is exceeded. This extends the limiting effect of soil water on $POD_1SPEC$ further north to substantial parts of the mid-latitudes.


Southern Europe shows the highest variation of $POD_1SPEC$ during the five years, most likely highlighting the influence of more pronounced changes in weather and hence ozone uptake conditions during those five years as compared to the rest of Europe. However, it should be noted that none of the five years targeted in this study was perceived as a meteorologically extreme, with none of the extreme European heat waves that characterised e.g. 2003 (Solberg et al., 2008, Lin et al., 2020).


The $POD_1SPEC$ results in the present study are are consistent with Anav et al (2022), who found that ~ 40 % of the forested area in the Northern Hemisphere exceeded the $POD_1SPEC$ values of critical levels used in their study (5.2 mmol m$^{-2}$ y$^{-1}$ for boreal and continental deciduous forests, 9.2 mmol m$^{-2}$ y$^{-1}$ for boreal and continental evergreen forests, 14 mmol m$^{-2}$ y$^{-1}$ for temperate deciduous forests and 47.3 mmol m$^{-2}$ y$^{-1}$ for temperate evergreen forests). Sicard et al (2020) found broad ranges

of $POD_1SPEC$ between 3 to >20 mmol $O_3$ m$^{-2}$ for 15 locations in France, Italy and Romania during 2018 and 2019. A study at 30 rural sites in Switzerland found ranges in $POD_1SPEC$ for beech of 12 to 25 mmol $O_3$ m$^{-2}$ (Braun et al., 2022) whilst our estimates are between 22 and 28 mmol $O_3$ m$^{-2}$. For European wide studies, Simpson et al. (2022) calculated $POD_{1\_IAM\_DF}$ (the integrated assessment model recommendation of UNECE (LRTAP, 2017), which is largely derived from the parameterization for beech) with a more recent and higher resolution (rv4.45, 0.3 degrees lon x 0.2 degrees lat) version of the

EMEP CTM. They also found highest POD values (of around 12-40 mmol $O_3$/m$^2$) in central Europe, Sweden and the UK, and lower values (0-6 mmol $O_3$/m$^2$) in many parts of Spain, rather similar to the results presented here. Finally, a recent study by Vlásáková et al (2024) estimated $POD_1SPEC$ for beech and Norway spruce according to the UNECE Mapping Manual





functions and the CAMS ensemble forecast modelling which provided hourly $O_3$ data. These results are consistent with our study findings, with $POD_1SPEC$ for beech commonly ranging from 5 to 30 mmol $m^{-2}$ in a similar spatial pattern to that

described by our broadleaf deciduous results in Figure 4c and for Norway spruce commonly ranging from 7 to 25 mmol $m^{-2}$ compared to our slightly lower values for coniferous forests of 5 to 18 mmol $m^{-2}$.

## 4.2 The GAI dose-response relationships

We have developed GAI DRR responses so that the impacts of $O_3$ exposure on forest stand growth can be estimated annually and used with commonly available forest statistics to estimate annual changes in forest standing stocks and ultimately living

biomass C stocks. A number of assumptions have been made in the derivation of these GAI DRRs. Firstly, we assume that $O_3$ effects on young forest trees will be equivalent to those effects on mature trees, at least in terms of the $O_3$ effect on growth rate. This is further discussed in section 4.4. Secondly, we assume that we can reliably estimate the growth rate for different tree species using the Richards equation [Eq. 5] (Richards, 1959). Given that the Richards equation is used to interpolate growth rates between very young trees (when biomass will be close to zero) and known biomass of trees at the end of a 2-to-

10-year growth cycle, the uncertainties in this assumption are very small.

## 4.3. Impacts of ozone exposure on the forest living biomass C stock changes

Estimates of $O_3$ impacts on forest C sequestration for the different European countries, as well as for Europe in total, were in general much larger when based on the difference between GAI and total removals, compared with estimates based on ozone impacts on GAI itself. The removal of $O_3$ exposure was estimated to increase European forest stem volume growth rates by

9%, but it was estimated to increase European forest living biomass C stock changes by 31%, i.e. more than three times as much. This illustrated the importance of not only considering ozone impacts on forest growth rates but also to consider the impacts on the gap between gross growth rates and forest removals, i.e. the net changes in the forest standing stocks. Previous studies of the impacts of $O_3$ on vegetation C sequestration have in most cases focussed on the impacts of $O_3$ on photosynthesis (Felzer et al., 2005; Felzer et al., 2009; Sitch et al., 2007; Ren et al., 2007). However, direct impacts of environmental changes

on C-sinks involved in the growth process has been shown to be more important for tree growth than indirect control via impacts on C-sources, such as photosynthesis (Millard et al., 2007; Fatichi et al, 2014; Körner, 2015; Eckes-Shephard et al., 2020; Kannenberg et al., 2022). This would suggest that approaches that estimate the negative impacts of $O_3$ on vegetation C sequestration only via impacts on instantaneous photosynthesis may not be correct and hence care should be taken when making estimates of ozone effects on forest growth using damage functions that only influence via photosynthetic based carbon

assimilation (e.g. Sorrentino et al., 2025). The dose-response relationships used in our study were based on $O_3$ exposure experiments with the long-term biomass growth as the response parameter, thus mainly reflecting $O_3$ impacts on the C-sinks.



## 4.4. The applications for forest stands

The phytotoxicity of $O_3$ for young trees under experimental conditions is well documented (Karlsson et al., 2007; Wittig et al., 2009), but negative impacts of ozone on mature trees under field conditions have been more difficult to establish (Kolb and

Matyssek, 2001; Marzuoli et al., 2020). The most evident and well described case of negative ozone impacts on forest ecosystems are the ozone induced decline in the San Bernadino Forest in California (Miller and McBride, 1999). Negative impacts on forest growth were analysed mainly based on gradient studies (Miller et al., 1997; Arbaugh et al., 1999). It was estimated that the basal area increment growth rates of the Douglas fir forest were reduced by approximately 30%, when comparing the areas with daylight mean $O_3$ concentration of 64-67 and 76-83 ppb, respectively. These are high concentration

values compared to the corresponding values that can be estimated in Europe today (Gaudel et al., 2018). On the other hand, the climate in San Bernadino Forest is a Mediterranean dry climate which might be expected to limit $O_3$ uptake.

The impacts of $O_3$ on the growth and vitality of different tree species have been studied extensively in Switzerland (Braun et al., 2014; 2017; 2022). These studies include both experimental exposure studies with young trees as well as epidemiological

field studies where the stem growth as well as other vitality parameters have been correlated with environmental factors, including $O_3$ exposure. The agreement of the dose–response curve from the epidemiological studies with the experimental data with young Beech and Norway spruce trees suggests that the current dose–response curves for $O_3$ used within the LRTAP convention are also valid for mature forests (Braun et al., 2022). In a unique case study, a free-air ozone fumigation experiment was conducted at ''Kranzberger Forst'' (Freising, southern Germany), over 8 years on adult trees of European beech and

Norway spruce. Pretzsch and Schütze (2018) concluded that in the twice ambient $O_3$ exposure the annual basal area growth of European Beech and Norway spruce decreased by 32 and 24%, respectively. Free air fumigation of different deciduous tree species in the Aspen-Face project (Karnosky et al., 2005; Talhem et al., 2014) demonstrated a negative impact of $O_3$ on the ecosystem C content that, however, had a transient development over time. Experimental studies in China (Feng et al., 2019) with poplar tree species demonstrated a consistent negative relationship between biomass accumulation and O3 exposure both

of small trees in open-top chambers and larger trees under chamber-less exposure. Oksanen et al. (2009) summarized the results from Finnish experiments with $O_3$ exposure of several different clones of Birch and concluded that an $O_3$ exposure of 15 ppm hours AOT40 may reduce the stem diameter growth of Birch by approximately 15 %. This is of the same magnitude as the estimates of AOT40 described for large parts of Europe (Figure 3F). Thus, based on literature results described above, mature trees under field conditions cannot be assumed to be less sensitive to ozone exposure compared to young trees under

experimental conditions (Karlsson et al., 2024). Consequently, we suggest that our estimates of negative impacts of $O_3$ on forest growth rates and C stock changes, using DRRs from 2- to 10-year-old forest trees under experimental conditions, may provide reliable results for the different forests of variable age that exist across Europe.





### 4.5. Assumptions

The estimates of the impacts of $O_3$ exposure on the C sequestration of the managed forests were based on some additional
assumptions. The first assumption was that the presence of $O_3$ did not affect the natural losses caused by storm fellings, drought,
insect attacks etc. Free-air fumigation to elevated concentrations of $O_3$ were shown to increase aphid infestations in trembling
aspen, *Populus tremuloides* (Percy et al., 2003). If absence of $O_3$ exposure would reduce the risk for insect attacks or other
diseases in trees, that would contribute to increase the C sequestration in European managed forests even further, since the gap
between gross growth and removals would increase. Ozone has been shown to reduce the relative growth of below ground
biomass (Gu et al., 2022). Reduced root growth can be expected to make trees more vulnerable to drought. Hence, the absence
of $O_3$ exposure would have the potential to increase the C sequestration also from this aspect.

The second assumption was that harvest rates would not be affected by the changes in gross growth rates, i.e. that harvest rates
are mainly dependent on the demand for forest raw materials. In countries that are heavily dependent on the supply of forest
raw material to the forest industry, such as e.g. Sweden and Finland, the rates of harvests are carefully monitored by national
forest inventories. For example, the Swedish Forest Agency regularly make recommendations about the rates of harvest in
relation to the rates of forest gross growth, to make sure that harvest rates do not exceed growth rates. Levers et al. (2014)
concluded that the harvest intensities of European forests were mainly explained by the share of plantation species, by terrain
ruggedness, and by different country-specific characteristics. Forest growth rates were not considered as important for
explaining forest harvest intensities in that study. Another important factor regulating the harvest rates is the forest age
structure, i.e. when forests reach the economically optimal age for harvesting (Korosuo et al., 2023). An alternative assumption
for the calculations would have been to keep harvest rates as a certain fraction of the growth rates. In this case increased growth
rates in the absence of $O_3$ exposure would have been accompanied by a certain increase in harvest rates, that in absolute terms
would have been lower than the increase in growth. Hence, also in this case, the absence of $O_3$ exposure would have resulted
in increased C sequestration in European managed forests, but it would have been smaller than what has been presented in the
current calculations in our study.

### 4.6. Comparisons with other estimates of and policy for European forest carbon (C) sequestration

In the present study, the yearly C sequestration to the European forests living biomass C stocks, as a mean value 2008 – 2012,
was estimated to be 343 M t $CO_{2e}$ yr$^{-1}$ in the presence of $O_3$. Regarding the uncertainties and simplification of the calculations,
this value is in the same order of magnitude as the value estimated in the Forest Europe assessment (Forest Europe, 2020), that
the average annual sequestration of C in European forest biomass between 2010 and 2020 was around 560 M t $CO_{2e}$ yr$^{-1}$. Pan
et al (2011) estimated the yearly C sequestration to European forests, excluding Russia, to 570 M t $CO_{2e}$ for the period 2002-
2007.



The European Union European Commission has issued an action plan to develop sustainable solutions to increase C sequestration by forests. The EU target is to increase the yearly uptake of $CO_2$ in the entire AFOLU sector (Agriculture, Forestry and Other Land Use) to 310 M t $CO_{2e}$ yr$^{-1}$ (Korosuo et al., 2023). It was estimated in the Forest Europe assessment (Forest Europe, 2020) that living woody biomass represents around 36% of the total C stocks in forests. Hence, the increase in the yearly C sequestration to the European forests living biomass C stocks estimated in this study, by eliminating negative

impacts of $O_3$ on forest gross growth rates, from 343 to 449 M t $CO_{2e}$ yr$^{-1}$, clearly has the potential to substantially contribute to the European Commission action plan. The yearly C sequestration to the European forests is expected to be short of the so called "forest reference levels", i.e. the forest C sequestration levels agreed by the EU member states during 2021–2025 (Hyyrynen et al., 2023; Korosuo et al., 2023).

The estimates made in this study included only the $O_3$ influence of changes in the forest living biomass C stocks. However, there may be additional climate change abatement benefits caused by mitigating ozone concentrations, such as enhancing C held in the dead biomass and soil C stocks (Liski et al., 2002; Mäkipää et al., 2023). Both these C stock changes may be promoted by the increase litter production due to the increase in forest gross growth rates. Furthermore, C can be stored for some time in harvested wood products (HWP, Hyyrynen et al., 2023) increasing C stocks. Finally, the forest raw materials

produced may substitute the use of fossil-based raw materials (Leskinen et al., 2018; Korosuo et al., 2023).

## 5 Conclusions

- The annual, species-specific Phytotoxic Ozone Dose above a threshold of 1.0 nmol m$^{-2}$ s$^{-1}$, POD$_1$SPEC, was estimated for European forests plant functional types for the years 2008 – 2012.
- The critical level for negative impacts on forests suggested by the UN Air Convention, based on POD$_1$SPEC, was

620           exceeded in large parts of Europe.
- The highest POD$_1$SPEC was estimated for the coastal regions of mid-latitude Europe including UK, for both coniferous and broadleaf forests.
- The annual forest standing stock changes were estimated based on POD$_1$SPEC -based dose-response relationships for negative impacts of ozone on forest growth rates in combination with the estimated values for POD$_1$SPEC as well as

625           official forest statistics for forest growth and total removals. With the use of default IPCC methodology, this could be converted to estimates of changes in the forest living biomass C stocks.
- The absence of ozone exposure would increase European forest stem volume growth rates by 9%, but it would increase European forest living biomass C stocks by 31%.
- Mature trees under field conditions cannot be assumed to be less sensitive to ozone exposure compared to young trees

630           under experimental conditions.



## 6 Author contribution

PEK, PB and LE designed the study. DS and SB performed the simulations. KS and FH provided important information. PEK, PB and LE prepared the manuscript with contributions from all co-authors.

## 7 Acknowledgements

The contribution by Per Erik Karlsson to this study was supported by the Swedish Environmental Protection Agency. Lisa Emberson and Sam Bland were partly supported by a grant (grant no. NE/V02020X/1) from the Future of UK Treescapes research program, funded by the UKRI. The work of David Simpson with the EMEP model was funded by the EMEP Trust find. IT infrastructure in general was available through the Norwegian Meteorological Institute (MET Norway), with EMEP computations performed on resources provided by UNINETT Sigma2 – the National Infrastructure for High Performance 640 Computing and Data Storage in Norway (grant nos. NN2890k and NS9005k). Thanks also to Gina Mills, Harry Harmens and David Norris (formerly of UKCEH Bangor) for support in developing the original idea for the study as well as to Sabine Braun, Institute for Applied Plant Biology AG, Witterswil, Switzerland, for valuable discussions. We would also like to acknowledge support in the application of the SEI land cover map from Steve Cinderby and Howard Cambridge (SEIY); and support in data analysis from Connie O'Neill (SEIY) and Nathan Booth (E&G Dept of University of York).

## 8 Data availability

Data will be available upon request.

## 9 Competing interests

The authors declare that they have no conflict of interest.

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
