# Peer review of "Ozone causes substantial reductions in the carbon sequestration of managed European forests"

_EGUsphere, 2024_

## Referee Comment (RC1)

Review: P.E. Karlsson et al., Ozone causes substantial reductions in the carbon sequestration of managed European forests. EGUsphere, Jan 2025.
https://doi.org/10.5194/egusphere-2024-3742
Received 3 Jan 2025
Reviewed 16 Jan 2025

**General Comments**

This work combines modelled ozone concentrations across Europe with a forest response model to quantify the effects of ozone damage, compared to a pre-industrial scenario, on carbon sequestration. The results show that these effects are substantial, especially if expressed as a fraction of the gap between gross stem volume growth and natural & harvest carbon removals. This has important implications not just regarding the sustainability of forestry in Europe, but obviously also for the role of forests in climate change mitigation, and provides another reason for supporting efforts to improve air quality.

In my opinion, possibly the most important sentence in the paper is on Line 469 in 4.1, stating that the exceedance maps derived using the methods in this paper are very different from those used by the EU, based on AOT40. This is worth repeating at the beginning and end of the paper.

The manuscript is well written and appears to be scientifically sound. Some of the explanations can be improved, and suggestions are given below.

**Specific Comments**

L170: that is not what is in LRTAP Ch. 3. Either explain the modifications applied to the LRTAP formulation or provide a more current reference.

L202: 1 nmol seems a bit arbitrary, particularly if it is applied to all species. Perhaps discuss in 1 or 2 sentences how this number has been justified previously, and whether adjusting it up or down makes much of a difference.

L326: verify whether you mean 3a, e and f (and not a, c and f). Define AOT40.

L398, Fig. 6. This figure could be improved significantly with some additional labels. I recommend putting the numbers right next to the bars (933, 954, 571, 363, and the difference of 80 between the first two) to facilitate making links to the text. I would also label the two arrows (e.g. a, b), which would let you be specific in the text to explain that a = 80 = 9%, b = 283, and a/b = 28%.

L405: This paragraph is somewhat confusing. Add a sentence (or replace L407-409) stating that the difference between SVSC due to harvest & natural removal is 283 with $O_3$, but 363 without, which is 28% higher. Change L408 to "This is a decrease by 28% in the presence of (industrial) $O_3$."

L418: The absolute difference

L419: a much larger relative impact

L468: dry areas in the Mediterranean; I think Iberia would be more precise

L616: This actually reads like a Summary, not a Conclusion

L628: as stated, this is incorrect – be precise: would increase sequestration to living biomass (i.e. a rate of change, not an absolute change)

Figures 5 and 8 and Table 2 could be moved to the SI to consolidate the manuscript for improved readability.

**Technical Comments**

L30: superfluous "and"
L87: is often
L170 (Eq [2]): missing )
L254, Fig. 2: label ANL on the figure
Fig. 3: to make this figure more readable: 1. Add labels to each panel (e.g. (a) accumulated conifers, (b) range (conifers), and 2. In (e) and (f), round the numbers on the color scale to nice integers
L333: change "from the" to "since "
Fig. 6. "ob" should be spelled out in the caption. Superfluous "-" after natural
Fig. 8. No reason for the y-axis in b) to go to 40; go to 20 to improve resolution
L484: overestimated
L486: at
L504: superfluous ); define rv

---

## Author Comment (AC1)

**Responses to referees comments, anonymous Referee, RC1**

Below, we address the different comments given by Anonymous Referee RC1. We provide our responses to the comments, describe the overall actions that we suggest as a response to the referee´s comments and finally in detail describe the text that we suggest to be removed or revised to a new, revised version of the manuscript.

The line numbers that we use in this document refer to the line numbers in the original pdf-file of the submitted manuscript as used by the referees.

**RC1 general comments**

"The manuscript is well written and appears to be scientifically sound. Some of the explanations can be improved, and suggestions are given below."

"In my opinion, possibly the most important sentence in the paper is on Line 469 in 4.1, stating that the exceedance maps derived using the methods in this paper are very different from those used by the EU, based on AOT40. This is worth repeating at the beginning and end of the paper."

"Figures 5 and 8 and Table 2 could be moved to the Supplementary Information (SI) to consolidate the manuscript for improved readability."

Authors responses.

We thank referee 1 for this positive overall judgement.

Also in the opinion of the authors, the result on the difference in the maps for exceedance based on POD and AOT40 is important. However, equally important is the result the ozone impacts on European forest carbon sequestration will be much larger when calculated based on the impacts on the gaps between gross growth and total removals, compared to when based only on the impacts on gross growth. In many modelling studies, the impact on gross growth is used, which results in an underestimation of ozone impacts on forest carbon sequestration.

We have been intending to stress both these two important results in the article.

Action taken

No action.

The referee suggests to move Figures 5 and 8 and Table 2 to the SI.

Figure 5 concerns an important improvement made in the current article, by transferring dose-response relationships based on impacts on the percent annual biomass loss used in the Mapping Manual to dose-response relationships for the estimated impacts on gross annual increment. Even though the dose-response did not change very much, this represents an important methodology improvement, since impacts on percent annual biomass loss can not be used in combination with forest statistics, which need information about impacts on gross annual increment rates.

We suggest that Figure 5 remains in the main text.

Figure 8 concerns changes in the annual C sequestration in the living biomass C stocks, with and without the exposure to present O3 doses. We consider this as an important figure, since it can be used as information for the individual European countries.

We suggest that Figure 8 remains in the main text.

Table 2 concerns the percent reductions in gross annual increment rates, caused by the present O3 exposure, used for the different European countries. This is important information in order to understand the size of the O3 impacts on the annual C sequestration for the different countries (Figure 8).

We suggest that Table 2 remains in the main text.

**RC1 specific points**

Line: 170

Referees comment.

That is not what is in LRTAP Ch. 3. Either explain the modifications applied to the LRTAP formulation or provide a more current reference.

Authors response.

Yes, the referee is correct.

This is what is in the Mapping Manual:

(III.1) $g_{sto}$ = $g_{max}$ * [min($f_{phen}$, $f_{O3}$)] * $f_{light}$ * max{$f_{min}$, ($f_{temp}$ * $f_{VPD}$ * $f_{SW}$)}

And this is what is used in the current article:

$$g_{sto} = g_{max} \cdot f_{phen} \cdot f_{light} \cdot max\{f_{min}, (f_{temp} \cdot f_{VPD} \cdot f_{PAW}\}$$

What is in the Mapping Manual is the more general equation that can be used for all different types of vegetation. The $f_{O3}$ is used only for agricultural crops, so it was not

included here. Furthermore, we replaced the $f_{SW}$ with $f_{PAW}$. The latter is interpreted as f "Plant Available Water" and is essentially the same as $f_{SW}$ ("Soil Water").

Action taken.

We added text to clarify this in the sentence starting on line 166.

Line: 202

Referees comment.

1 nmol seems a bit arbitrary, particularly if it is applied to all species. Perhaps discuss in 1 or 2 sentences how this number has been justified previously, and whether adjusting it up or down makes much of a difference.

Authors response.

A main strategy for this article was to apply the methods for calculating POD1SPEC that have been agreed in the LRTAP convention ICP Vegetation Mapping Manual for different tree species across Europe and then explore the consequences of the estimated PODySPEC, in combination with the dose – response relationships also accepted in the Mapping Manual, for the carbon sequestration by European forests. In that context, we did not want to modify the methods from the Mapping Manual.

However, the comment by the reviewer is interesting and in fact, one of the authors (PEK) has explored the consequences for using alternative values for the threshold value for possible ozone impacts on forests in Sweden. It turned out that the choice of threshold value had relatively small influence.

Reference: Karlsson. P.E., Danielsson, H., Pleijel, H., Andersson, C. 2024. Exceedance of critical levels for ozone impacts on Swedish forests - Evaluation of methodology for POD1SPEC calculations. IVL Report C 829.

However, as the aim of this study was to apply, rather than discuss the methods used in the Mapping Manual to calculate POD1SPEC we have not included further discussion of this issue but have clarified that the use of POD1SPEC is consistent with the metric recommended by the UNECE Mapping Manual (LRTAP, 2017) on line 140.

Action taken.

We added some text to clarify this.

Line: 326

Referees comment.

Verify whether you mean 3a, e and f (and not a, c and f). Define AOT40.

Authors response.

The following relevant panels are included in Figure 3:

3a, POD1SPEC for coniferous species

3c, POD1SPEC for broadleaf species

3e, AOT40, Norway Spruce

3f, AOT40, Beech

So, in fact all these four panels are relevant when comparing the geographical distribution of the estimated values for POD1SPEC and AOT40, respectively.

So, thank you to the referee for spotting this error.

Action taken.

The text was revised.

Text added or removed

New text: Fig. 3a, c, e and f

Line: 398, Fig. 6.

Referees comment.

This figure could be improved significantly with some additional labels. I recommend putting the numbers right next to the bars (933, 954, 571, 363, and the difference of 80 between the first two) to facilitate making links to the text. I would also label the two arrows (e.g. a, b), which would let you be specific in the text to explain that a = 80 = 9%, b = 283, and a/b = 28%.

Authors response.

We agree that the suggestions made by the referee might facilitate understanding the results presented in the figure 6. However, in general journals recommend that diagrams should not contain too many symbols within the frames of the diagrams. The authors could however not find a recommendation of this with the Authors instructions of Biogeosciences. Furthermore, we face the risk that the diagram could be confusing if there are too many symbols inside the diagram frame. And, as the referee also states, all these values are available in the main text.

Action taken.

We will consult the editor about advice on this matter. If the editor agrees, we will change the Figure 6 according to the suggestions made by the referee.

Line: 405

Referees comment.

This paragraph is somewhat confusing. Add a sentence (or replace L407-409) stating that the difference between SVSC due to harvest & natural removal is 283 with $O_3$, but 363 without, which is 28% higher. Change L408 to "This is a decrease by 28% in the presence of (industrial) $O_3$."

Authors response.

The authors have worked very hard with this paragraph in order to make it as clear and logical as possible. First, we state the annual increase gross stem volume increment in the presence of $O_3$ and then in the absence of $O_3$ and finally the difference between these values in percent (+9 %). After that we state the annual increase in the forest standing stocks in the presence of $O_3$ and then in the absence of $O_3$ and we then state the difference between these value (+28 %). We then conclude with summarizing the difference between the % increases caused by the absence of $O_3$, +9 % in the case of annual increase gross stem volume increment and +28 % in the case of the annual increase in the forest standing stocks.

We appreciate the suggestion by referee 1, but as we were quite satisfied with the original text, we would suggest not to change the original text.

Action taken.

No action

Line: 418

Referees comment.

The absolute difference

Authors response.

Agree.

Action taken.

Text corrected

Text added or removed

The word "absolute" was added in line 418

Line: 419

Referees comment.

a much larger relative impact

Authors response.

Agree

Action taken.

Text corrected

Text added or removed

The word "relative" was added in line 418

Line: 468

Referees comment.

dry areas in the Mediterranean; I think Iberia would be more precise

Authors response.

Agree

Action taken.

Text changed

Text added or removed

"Mediterranean" was changes to "the Iberian peninsula"

Line: 616

Referees comment.

This actually reads like a Summary, not a Conclusion

Authors response.

We do not quite agree with the referee, we regard this as the main conclusion from the work presented in the article.

Action taken.

No action

Line: 628

Referees comment.

as stated, this is incorrect – be precise: would increase sequestration to living biomass (i.e. a rate of change, not an absolute change)

Authors response.

Agree.

Action taken.

Text corrected

Text added or removed

From "The absence of ozone exposure would increase European forest stem volume growth rates by 9%, but it would increase European forest living biomass C stocks by 31%."

To "The absence of ozone exposure would increase European forest stem volume growth rates by 9%, but it would increase European forest living biomass C stocks increment rates by 31%."

**RC1, Technical Comments:**

Line: 30

Referees comment.

superfluous "and"

Authors response.

Corrected

Line: 87

Referees comment.

is often

Authors response.

Corrected

Line: 170

Referees comment.

(Eq [2]): missing )

Authors response.

We do not understand this comment, equation is already labelled [2]

Action taken.

No action

Line: 254

Referees comment.

Fig. 2: label ANL on the figure

Authors response.

OK, includes also ATF.

Action taken.

ANL as well as ATF was added to Figure 2.

Line: Fig. 3:

Referees comment.

to make this figure more readable: 1. Add labels to each panel (e.g. (a) accumulated conifers, (b) range (conifers), and 2. In (e) and (f), round the numbers on the color scale to nice integers

Authors response.

OK, figure 3 will be modified according to the referee´s suggestions.

Line: 333

Referees comment.

change "from the" to "since "

Authors response.

OK

Text added or removed

changed "from" to "since "

Line: Fig. 6.

Referees comment.

"ob" should be spelled out in the caption. Superfluous "-" after natural

Authors response.

OK

Action taken.

"ob" should be spelled out in the caption, "-" removed in the figure

Line: Fig. 8.

Referees comment.

No reason for the y-axis in b) to go to 40; go to 20 to improve resolution

Authors response.

We have to make some space for the legend, but we suggest the y-axis to go to 30

Action taken.

Figure 8b modified

Line: 484

Referees comment.

overestimated

Authors response.

OK

Action taken.

Corrected

Line: 486

Referees comment.

at

Authors response.

Corrected

Line: 504

Referees comment.

superfluous ); define rv

Authors response.

OK

Action taken.

")" removed, text added to define "rv"

---

## Author Comment (AC2)

**Responses to referees comments, Anonymous Referee #2**

Below, we address the different comments given by Anonymous Referee #2. We provide our responses to the comments, describe the overall actions that we suggest as a response to the referees comments and finally in detail describe the text that we suggest to be removed or revised to a new, revised version of the manuscript.

The line numbers that we use in this document refer to the line numbers in the original pdf-file of the submitted manuscript as used by the referees.

**Referee #2 general comments**

"Karlsson and others estimate annual ozone uptake by European forests. This is important work and I feel that the manuscript is meritous but could be further improved in key areas."

Authors response.

We thank referee 2 for this positive overall judgement.

Referees comment.

The abstract wanders a bit and I'm not entirely sure what statements like 'limited to the north by mid-Sweden and south Norway and Finland' mean in practice.

Authors response.

With this sentence we are trying to describe the results presented in the maps of Figure 3a and c. It is difficult the describe the geographical pattern of the highest POD in just a few words. This was the best that we could describe the results in this respect.

Action taken.

The text in the abstract has been modified to remove over wordy sentences improving clarity of the text.

Referees comment.

I don't have too many critiques of the results and workflow because I feel that it is state of the art (and the authors know how models can be improved) and other suggestions would likely be arbitrary and not meaningfully impact final results.

Authors response.

Thank you.

**Referee 2 specific points:**

Line: 29

Referees comment.

'as well as for and the'. Note minor usage errors like this throughout the manuscript

Authors response.

Thank you for spotting this error!

Action taken.

Corrected

Text added or removed

Text corrected on line 29 (and in other parts of the manuscript)

Line: 56 and 59

Referee´s comment.

56: '4.000 Mt CO2e' has a remarkably precise number of significant digits. See also Line 59.
'560 M t CO2e' is more believable.

Authors response.

This was not intended as a precise number, this was an error, the point was used as
"separator" to represent "four thousand."

Action taken.

Corected

Text added or removed

Changed to "4 000"

Line: 63

Referees comment.

Please note that 'C sequestration of managed forests to a large extent depends on the
balance between forest growth and removals' refers only to the site scale because the full

carbon accounting will depend on the ultimate use of the wood in relatively long-lived stocks or short-lived ones like bioenergy

Authors response.

We agree with the referee. However, this study was limited to living biomass carbon stock changes in the forest ecosystems. The methods for carbon accounting calculations the carbon in long lived harvested wood products are still debated. Even more uncertain are calculations of substitution effects. We mention this is the discussion section.

Action taken.

Clarified.

Text added or removed

Text modified: "The sentence starting on line 62 was changed to: "In general, forests that are actively managed sequester C in the forest ecosystem carbon stocks at higher rates than non-managed forests (Nabuurs et al., 2008)."

Line: 68

Referees comment.

'that has been found to cause substantial losses to' -> 'causes losses' because the word substantial is subjective and there is causality in this case.

Authors response.

OK, we agree.

Action taken.

Corrected

Text added or removed

The word "substantial" was removed.

Line: 121

Referees comment.

This is incorrect. The atmospheric surface layer i.e. planetary boundary layer during most daytime conditions will be on the order of hundreds of meters to kilometers

Authors response.

The surface layer (SL, also known as the constant flux layer) and planetary boundary layer (PBL) are quite different entities, and we think we have used the term surface layer in its normal meteorolgical meaning. For example, the American Meteorological Society (https://glossary.ametsoc.org/wiki/Surface_boundary_layer) defines the SL as follows: A layer of air of order tens of meters thick adjacent to the ground where mechanical (shear) generation of turbulence exceeds buoyant generation or consumption. In this layer Monin–Obukhov similarity theory can be used to describe the logarithmic wind profile. The friction velocity u* is nearly constant with height in the surface layer. The exact height of the SL is difficult to define, but is typically thought to be about 4-10% of the PBL height (Stull, 1988, Pielke, 2002).

Pielke, R. A., Avissar, R., Raupach, M., Dolman, A. J., Zeng, X. B., and Denning, A. S.: Interactions between the atmosphere and terrestrial ecosystems: influence on weather and climate, Global Change Biology, 4, 461–475, 1998.

Stull, R. B.: An introduction to Atmospheric Boundary Layer Meteorology, Kluwer Academic Publishers, Dordrecht, 1988.

Action taken.

We have clarified L121 with a few extra words

Text added or removed

from: assumed to be the top of the atmospheric surface layer

to: assumed to be the top of the atmospheric surface, or constant flux, layer. cf Stull, 1988:

Line: 252

Referees comment.

I feel that fig. 2 can be smaller / more efficient (unnecessary white space and italicization)

Authors response.

OK, we changed figure 2, making it smaller and removing italicization

Action taken.

We changed figure 2, making it smaller and removing italicization

Text added or removed

Figure 2 was revised according to the suggestions made by the referee.

Line: 279

Referees comment.

Please don't use the * symbol in place of multiplication in formal equations. Also, this equation should be labeled.

Authors response.

OK

Action taken.

Corrected.

Text added or removed

Changed to the symbol "·". Equation was labeled.

Line: 296, Fig. 3

Referees comment.

Not sure if a map like this will pass colorblindness criteria

Authors response.

The map was checked versus colorblindness criteria

Action taken.

Figure 3 will be checked against colorblindness criteria and modified accordingly.

Line: 314

Referees comment.

Still not sure what 'limited to the north' means because there is a gradient and forests begin to dwindle of course once one gets too far north

Authors response.

In figure 3a and c, mean values for PODySPEC are represented in maps over Europe. Starting line 314 we try to in words describe the geographical distribution of high values of POD over Europe. We think that this description should not be too detailed, since the reader can see the maps by themselves, but we try to describe this large scale pattern for the high values. In that context, the authors think that there is a relatively clear limitation of high values of POD towards northern latitudes, "limited to the north by mid-Sweden and south

Norway and Finland". We do not think that we can clarify this further, without adding quite a lot of additional text. Again, the reader can see the distribution by themselves.

Action taken.

No action.

Line: 314, figure 3.

Referees comment.

The small font and poor quality of text in the colorbars needs to be improved for visibility (applies to a few figures)

Authors response.

OK.

Action taken.

The maps in the figures were corrected with respect to small font and poor quality of text in the colorbars.

Text added or removed

Revised maps added, also other maps were revised regarding small font and colorbars..

Line: 396, Fig. 6

Referees comment.

Fig. 6 would benefit from uncertainty bars

Authors response.

Figure 6 shows the rates of total, annual gross stem volume increment growth for forests in Europe as mean annual values for the time period 2008-2012. We assume that the referee suggest that we add uncertainty bars representing the variability between the five different years 2008 – 2012. Unfortunately this is not possible since forest statistics are only provided as the mean values for the five year period. In fact, this is generally the tradition for forestry statistics, to be provided as running 5-year mean values.

Action taken.

No action

Line: 410

Referees comment.

It would be good to point out here and elsewhere that these are the results that follow from the assumptions of the models

Authors response.

OK.

Action taken.

We tried to clarify this by adding text.

Text added or removed

Text added to Line 410 to replace the word "This": "The results from this study, with modelling of negative ozone impacts on forest growth in combination with the application of forest statistics,"

---

## Author Comment (AC3)

**Responses to community comments, CC1**

Below, we address the different comments given by community comments #1. We provide our responses to the comments, describe the overall actions that we suggest as a response to the referees comments and finally in detail describe the text that we suggest to be removed or revised to a new, revised version of the manuscript.

The line numbers that we use in this document refer to the line numbers in the original pdf-file of the submitted manuscript as used by the referees.

**CC1, general comments**

Referee comment.

"Interesting study, I am not an expert on this specific field, but I would suggest to compare your estimates, on section 4.6, with GHGI data submitted by EU MS (plus, in your case, Norway, UK and Swiss), rather than referring to quite old estimates (even if certainly important), such as Pan et al or, quite generic, like Forest Europe 2020. As far as I see, your estimates are well aligned with EU GHGI data submitted to IPCC. Indeed, looking to the data reported by CRF table 4.A for EU (National Inventory Submission 2023), for the period 2008-2012, the average C stock change in living biomass, in including UK, is around -362 Mt $CO_2$ yr-1. This value is fully consistent with your data (I haven't check for Switzerland and Norway, but this should not move too far the average for the same period). On the opposite, Pan et al and Forest Europe report quite different estimates (560 and 570 Mt $CO_2$), also referred to different periods, compared with your study. So. I would not consider them ….".

Authors responses.

We thank for this positive overall judgement. And thank for making the comparison with the CRF table 4.A for the EU. Good that there is a reasonable agreement. CRF table 4.A have a complicated structure with one sheet for each year, and it would be quite laborious to compile all these data for all European countries. This is outside the scope of this study.

Action taken

No action.

Referee comment.

On the same section, you also mention the EU target for LULUCF, equal to -310 Mt $CO_2$ yr-1. However, by comparing this value with your estimates referring to the period 2008-2012, you should even mention that the current sink reported by EU, is quite far from the value reported for 2008-2012, and even more far from the EU LULUCF target (as already highlighted by Korosuo et al., 2023). Indeed, looking again to the data submitted by EU in 2023, for the living biomass pool, the latest biomass sink reported (i.e. 2021) is around -230

Mt CO2 yr-1. So, the potential contribution of ozone to the biomass C uptake is certainly relevant but the current situation is quite different from 2008-2012.

Authors responses.

Again, thank for your analysis. As the referee quotes, the current sink reported by EU is far from the EU LULUCF target. We already mention this in the discussion, starting line 606 "The yearly C sequestration to the European forests is expected to be short of the so called "forest reference levels", i.e. the forest C sequestration levels agreed by the EU member states during 2021–2025 (Hyyrynen et al., 2023; Korosuo et al., 2023)." We do not think that we need to further stress that the current situation for EU is quite different from 2008-2012. We need to keep in mind that we in our study estimate only changes in living biomass carbon stocks, which is only a part to the total LULUCF.

Action taken

No action.

Referee comment.

Finally, I would suggest reporting all values referred to the biomass sink with negative sign, (i.e. -310 Mt CO2 yr-1, etc.), from an atmosphere perspective, in line with conventional approach used by IPCC.

Action taken

This is a good point, we will make this change throughout the article.